# Direct haplotype-resolved 5-base HiFi sequencing for genome-wide profiling of hypermethylation outliers in a rare disease cohort

Warren A. Cheung [1], Adam F. Johnson[1], William J. Rowell [2], Emily Farrow[1,3], Richard Hall [2], Ana S. A. Cohen[3,4], John C. Means[1], Tricia N. Zion[1], Daniel M. Portik[2], Christopher T. Saunders[2], Boryana Koseva[1], Chengpeng Bi[1], Tina K. Truong[5], Carl Schwendinger-Schreck[1], Byunggil Yoo[1], Jeffrey J. Johnston [1], Margaret Gibson[1], Gilad Evrony[5], William B. Rizzo[6], Isabelle Thiffault [3,4], Scott T. Younger [1,3], Tom Curran [7], Aaron M. Wenger[2], Elin Grundberg [1,3] ✉ & Tomi Pastinen [1,3] ✉

Long-read HiFi genome sequencing allows for accurate detection and direct phasing of single nucleotide variants, indels, and structural variants. Recent algorithmic development enables simultaneous detection of CpG methylation for analysis of regulatory element activity directly in HiFi reads. We present a comprehensive haplotype resolved 5-base HiFi genome sequencing dataset from a rare disease cohort of 276 samples in 152 families to identify rare (~0.5%) hypermethylation events. We find that 80% of these events are allele-specific and predicted to cause loss of regulatory element activity. We demonstrate heritability of extreme hypermethylation including rare *cis* variants associated with short (~200 bp) and large hypermethylation events (>1 kb), respectively. We identify repeat expansions in proximal promoters predicting allelic gene silencing via hypermethylation and demonstrate allelic transcriptional events downstream. On average 30–40 rare hypermethylation tiles overlap rare disease genes per patient, providing indications for variation prioritization including a previously undiagnosed pathogenic allele in *DIP2B* causing global developmental delay. We propose that use of HiFi genome sequencing in unsolved rare disease cases will allow detection of unconventional diseases alleles due to loss of regulatory element activity.

Short-read exome (srES) or genome (srGS) sequencing is the tool of choice for the detection of single nucleotide variants (SNVs) in most human genetic applications. However, even with strict case selection for rare genetic disease in a clinical trial, srGS achieved only a ~30% diagnostic rate[1], leaving most rare disease cases unsolved. On the other hand, 3rd generation long-read platforms, such as PacBio's

Single Molecule, Real-Time (SMRT) HiFi-GS technology generating 12–16 kb reads, have been demonstrated to produce not only high-quality SNV calls in difficult-to-map regions but also to accurately detect structural variants (SVs) genome-wide[2].

Within our large pediatric rare disease program, Genomic Answers for Kids (GA4K), collecting genomic data and health information for

families with a suspected genetic disorder, we have integrated an enhanced sequencing pipeline using HiFi-GS into the routine follow-up of unsolved cases. Our pilot phase of ~1000 families revealed that incorporating SVs from HiFi-GS resulted in new diagnoses in up to 13% of previously unsolved cases and HiFi-GS increased the discovery rate of rare coding SVs >fourfold compared with srGS[3]. However, the relative impact on undiagnosed rare diseases due to incomplete interpretation of detected variants such as those mapping to noncoding regions remains unknown.

We and others have shown that noncoding SNVs can have a pervasive effect on genome function and regulatory element (RE) activity, from splicing variation[4–6] to transcript levels in tissues and primary cells[7–9], as well as chromatin-states[10,11] and CpG methylation (mCpG) levels[12–14]. We exploited heterozygosity of functional alleles to augment the detection of differential RE activity[15] in the case of rare disease and for common population SNVs. We have also quantified other allele-silencing effects, such as imprinting or X-inactivation that are measurable in chromatin, mCpG and gene expression data[16–18]. We also previously showed that allelic RE hyper-mCpG reflects allelic regulatory/gene silencing and that genetic effects altering mCpG are more likely to be shared across tissues[19,20].

Established hyper-mCpG signatures are linked to several monogenic diseases[21,22] including imprinting disorders[23] and disorders caused by defects in chromatin regulators[24]. In addition, "epimutations" have emerged through locus-specific investigations, where a subset of missing rare disease alleles were shown to be noncoding and lead to hyper-mCpG and promoter inactivation[25]. In most known cases of disease-impacting hyper-mCpG the effect is restricted to one allele, but even higher resolution techniques using short-reads for mCpG (whole-genome bisulfite sequencing, WGBS)[20] lack resolution for genome-wide analysis of allele-specific effects.

The technical ability to interrogate mCpG status from single-molecular real-time (SMRT) sequencing (kinetic) data was recently demonstrated using the HiFi-GS long-read platforms[26]. This ability has now been integrated in the HiFi-GS production pipeline (Sequel IIe system release v11, PacBio, Menlo Park, CA) that combines a deep-learning model integrating sequencing kinetics and base context, to generate mCpG profiles genome-wide from standard sequencing libraries. The augmented 5-base HiFi-GS platform allows single-molecular resolution of mCpG together with phasing from long contiguous accurate reads, with the potential to detect allele-specific events at an order of magnitude increased efficiency as compared to WGBS. Using the throughput of Sequel IIe system here, we are able apply at scale the 5mC capability to interrogate fine resolution differences in methylation profiles from blood to interpret disease-relevant variation.

In this work, we leverage the large GA4K rare disease program and generate a comprehensive 5-base HiFi-GS dataset from 276 enrolled participants for parallel variant and mCpG calling. We validate the method through comparisons with WGBS and show high concordance in mCpG. We use the 5-base HiFi-GS dataset to identify >50,000 rare hyper-mCpG event that are to a large extent (80%) allele-specific and predicted to cause loss of RE activity (LREA). We exemplify the power of 5-base HiFi-GS in unsolved rare disease cases by identifying a hyper-mCpG event that caused LREA and led to a previously undiagnosed pathogenic allele in *DIP2B* causing global developmental delay.

## Results

### Benchmarking HiFi-GS for parallel genome and methylome assessments in a rare disease cohort

We used our large GA4K cohort[3] and generated a comprehensive mCpG dataset from 1367 enrolled participants utilizing WGBS (*N* = 1184) and 5-base HiFi-GS (*N* = 276) in peripheral whole-blood samples (Supplementary Data 1). Specifically, 1091 participants were

profiled by WGBS only, 183 by 5-base HiFi-GS only, and 93 by both platforms, respectively (Supplementary Data 2). As a first validation step of the integrated 5-base HiFi-GS pipeline for simultaneous mCpG profiling and GS, we assessed the effect of technical variability on mCpG profiles by comparing the 93 samples profiled by both WGBS and 5-base HiFi-GS. We extracted 16.9 million paired CpGs sites (>20× coverage) for sample-wise correlations and found remarkable consistency across the two methods (median Pearson *R* = 0.90, Supplementary Data 3). In addition, we evaluated the accuracy of mCpG profiling by extracting the top 500 most variable autosomal CpGs based on the HiFi-GS samples (Supplementary Data 2) for CpG-based correlations. Similarly, we noted high correlations compared to randomly permuted values (median Pearson *R* = 0.86; Supplementary Fig. 1; "Methods").

Next, we used individuals with unstable repeat disorders included in our dataset as positive controls to test how genome-wide mCpG data by WGBS performed at known disease-relevant regions. Specifically, we selected enrolled participants with *FXN* (Fig. 1A) and *FMR1* (Fig. 1B) mutations causing Friedreich ataxia (FRDA1)[27] and Fragile X syndrome[28], respectively, including two newly identified *FXN* intron 1 expansion carriers (Fig. 1A). While WGBS accurately identified hyper-mCpGs footprints in all carriers, the approach was not capable of reading into the pathogenic repeat or resolve carrier mCpG from homozygous sample. To contrast, we then applied 5-base HiFi-GS on additional probands with unstable repeat disorders, including a case with a *DMPK* repeat expansion causing congenital myotonic dystrophy type 1 (DM1)[22]. Here, haplotype-resolved HiFi-GS coupled with simultaneous high-resolution mCpG profiling allowed us to not only detect the *DMPK* repeat expansion but also its associated large (~1 kb) hyper-mCpG signature (Fig. 2).

Finally, we further confirmed[26] that haplotype-resolved mCpG across differentially methylated regions (DMRs) can be broadly utilized for the diagnosis of imprinting disorders[29]. We exemplified this resolution in unaffected individuals at known imprinted regions where HiFi-GS resolved the parent of origin of the allele-specific methylation pattern. Specifically, we exemplified this resolution in causal regions linked to Albright hereditary osteodystrophy[30] (*GNAS-AS1*; Supplementary Fig. 2), transient neonatal diabetes mellitus[31] (*PLAGL1*; Supplementary Fig. 3), Schaaf-Yang syndrome[32] (*MAGEL2;* Supplementary Fig. 4) and Temple Syndrome[33] (*MEG3/DLK1*; Supplementary Fig. 5).

### Identification and characterization of rare hypermethylation outlier event in rare disease cases

As shown for unstable repeat disorders, hyper-CpG can induce transcriptional silencing of disease genes via LREA. Thus, we hypothesized that screening for outlying hyper-mCpG events genome-wide may aid the identification of coding and noncoding, functional rare SNVs and SVs in unsolved rare disease cases to ultimately improve the diagnosis rate. We defined extreme hyper-mCpG outliers in two ways (Fig. 3). First, we ranked each CpG per individual based on the mCpG distribution for the population to catalog extreme hyper-mCpGs (Methods) and then classified hyper-CpG tiles (200 bp) if two or more hyper-CpGs in the tile were extreme. The hyper-CpG tile was further classified as rare (i.e., outlier) if it was only present in two or less unrelated individuals. Rare hyper-CpG tiles were then filtered reporting only those where the average z-score of all the CpGs of the tile was two or more. Using these criteria, we found a total of 25,543 extreme hyper-mCpGs tiles (Supplementary Data 4) with on average 125 hyper-mCpGs tiles per individual (Supplementary Data 5). The quantity of extreme hyper-mCpGs tiles per individual did not differ among unaffected and affected after taking sequencing coverage into account.

However, these conservative measures of extreme hyper-mCpGs tiles reduce true discovery at dynamic regions where mCpGs show high degree of variation in the population (Fig. 3). To extract unusual observations among such dynamic mCpGs tiles, we allowed

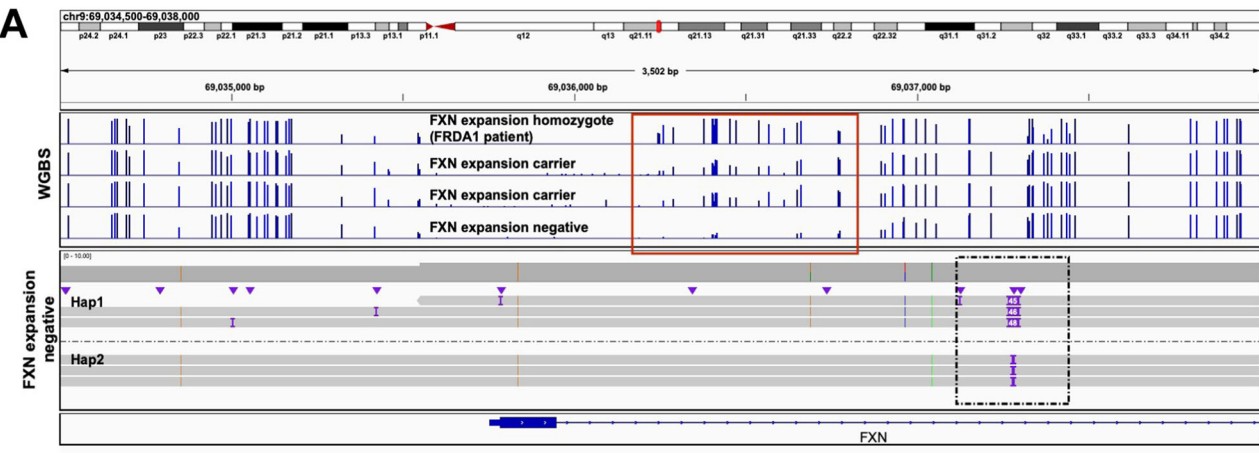

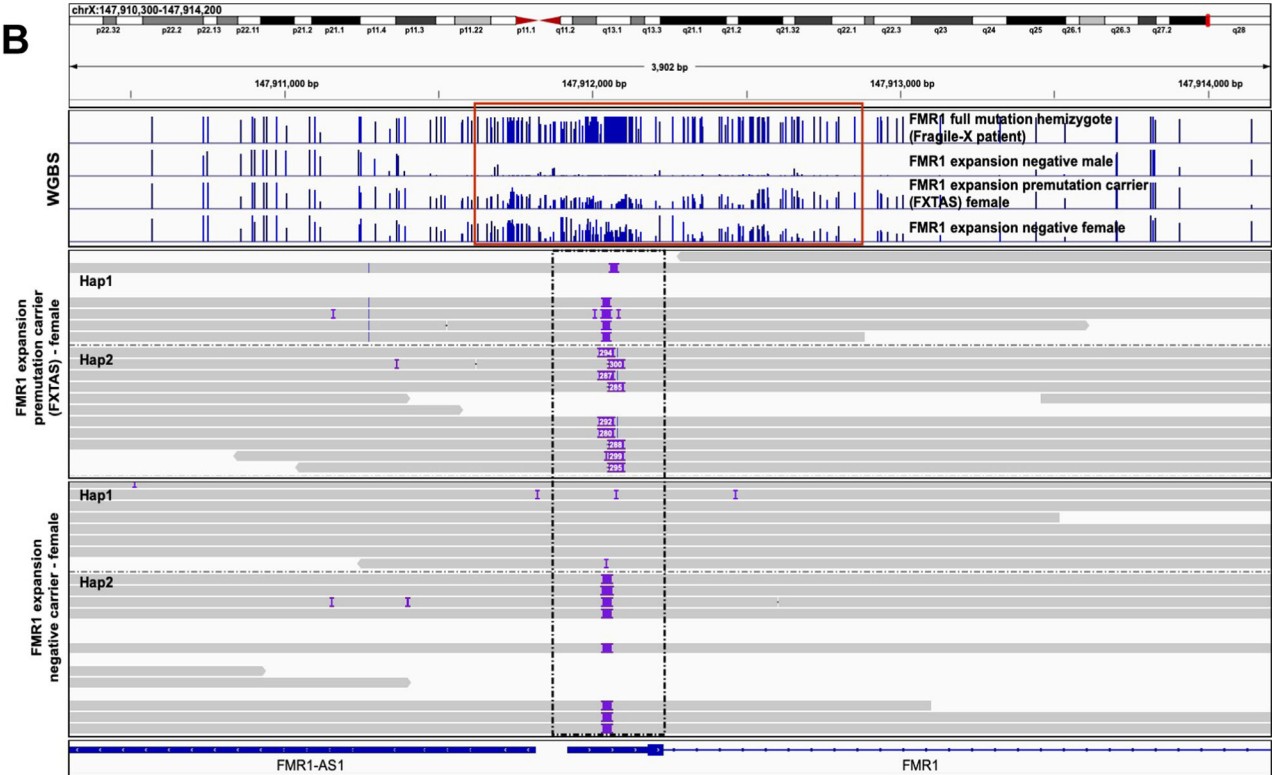

**Fig. 1 | Whole-genome bisulfite sequencing (WGBS) and HiFi-genome sequencing (HiFi-GS) in patients with unstable repeat disorders. A** Genomic view of 2.7 kb on chromosome 9 at the *FXN* locus showing one individual homozygous for the FXN expansion at pathogenic range (>65 repeats) causing the autosomal recessive disorder Friedrich's Ataxia (FRDA1), two FXN expansion carriers, and one individual without the expansion at a pathogenic range (<65 repeats, FXN expansion-negative). Blue tracks show the methylation level of CpGs measured by WGBS (*y* axis, 0–100%) with hypermethylation footprint (red box) linked to pathogenic repeat expansion (>65 repeats). Gray tracks show haplotype-resolved repeat expansion by HiFi-GS (black box, dashed line) in an individual without the

expansion at a pathogenic range (<65 repeats, FXN expansion-negative). **B** Genomic view of 3.7 kb on X chromosome at the *FMR1* locus showing an hemizygote individual with Fragile X (>200 repeats), one female FMR1 premutation carrier (55–200 repeats), and two individuals without the FMR1 expansion (male and female, <55 repeats). Blue tracks show the methylation level of CpGs measured by WGBS (*y* axis, 0–100%) with hypermethylation footprint (red box) linked to pathogenic repeat expansion (>200 repeats). Gray tracks show haplotype-resolved FMR1 repeat expansion (black box, dashed line) by HiFi-GS in one premutation carrier and one expansion-negative female. Hap1 denotes haplotype 1 and Hap2 denotes haplotype 2.

hyper-mCpGs tiles to be present in three or more unrelated individuals (removed the rarity criteria), but only kept instances where the average z-score of the CpGs in the hyper-mCpG tile was five or greater and with a minimum separation of three z-score units across unrelated samples allowing only a maximum of two samples per tile. These additional criteria retained another 31,005 hyper-mCpGs unique tiles

(Supplementary Data 6) and on average an additional 113 extreme hyper-mCpGs per individual (Supplementary Data 7).

Using the combined set of 56,548 unique hyper-mCpG tiles, we focused our analyses on the HiFi-GS subset carried out in affected patients (173 patients from 152 families; Supplementary Data 1) to provide the genome-wide characterization of rare hyper-mCpG events

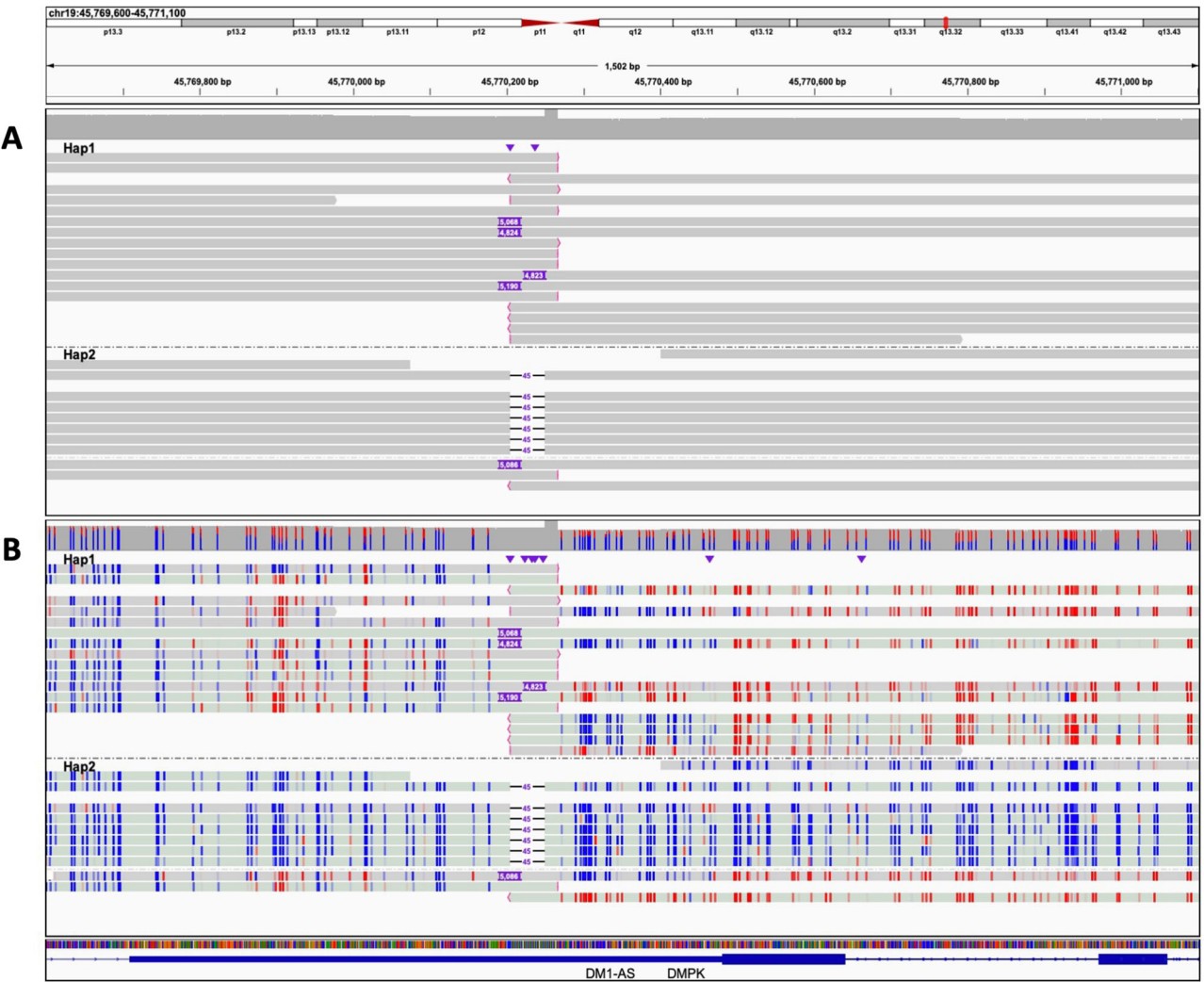

**Fig. 2 | HiFi-genome sequencing (GS) in a myotonic dystrophy type 1 (DM1) patient with DMPK repeat.** Genomic view of 1.5 kb on chromosome 19 at the *DMPK* locus showing an individual with the autosomal dominant disorder DM1. **A** Haplotype-resolved raw HiFi-GS reads identifying "soft-clipped" reads (i.e., reads with red ends indicative of sequence not matching the reference) and variable-sized insertion (4823–5190 bp) on haplotype 1 (Hap2) and a 45 bp deletion on haplotype 2 (Hap2). **B** Haplotype-resolved HiFi-GS reads with CpG modification coloring (blue indicating unmethylated CpG prediction and red indicating methylated CpG prediction with color intensity corresponding to base modification probabilities). Soft-clipped and insertion (expansion) spanning reads show consistent hypermethylation flanking the disease allele.

in a rare disease cohort. This restricted the set of rare hyper-mCpG tiles to 30,672 hyper-mCpGs (Supplementary Data 8). We assessed the haploid dependency of the extreme hyper-mCpGs (i.e., the outlier effect being observed on one allele only), which yielded 80% allele-specificity (Supplementary Data 8).

To further assess potential genetic underpinnings of rare hyper-mCpG tiles, we estimated the degree of sharing of the same hyper-mCpGs tile among related patients and observed a 3.1-fold enrichment ($P < 0.05$; "Methods"). Next, we queried the closest SNV (minor allele frequency, MAF, <0.5%) and SVs (MAF < 1%) for each hyper-mCpG tile and repeated the same test for all other mCpG tiles as a comparison. Using an empirical q value of less than 10% ("Methods"), we found that causal rare SNVs and rare SVs linked to the hyper-mCpG tiles were located within ~1–2 kb and within 10 kb for SNVs and SVs, respectively (Fig. 4A, B). To bolster the potential causality for local rare genetic variation, we exploited the haploid subset of hyper-mCpG tiles coupled with phased rare SNVs from the same reads (Fig. 4C) and showed statistically significant enrichment for local rare *cis*-SNVs extending up to 1 kb (Fig. 4D) from the hyper-CpG tile. However, we noted the strongest enrichment for SNV mapping close (<200 bp) to the hyper-mCpG bin

(Fig. 4E). Although WGBS were not able to resolve SNVs, it accurately detected associated hyper-mCpG footprints in probands carrying the rare SNVs (Supplementary Figs. 6–8). However, for local SVs and insertion (INS)-deletions (DELs), variations in reads are not phased ("Methods") and required manual curation. For 40 randomly chosen unphased short DELs called by DeepVariant and verified in read data and mapping near hyper-mCpG tiles, we observed 31 out of the 40 (binomial $P = 0.0007$) in *cis* suggesting similar behavior as for SNVs with respect to allelic and distance distribution (Supplementary Fig. 9).

Finally, most outlier effects were restricted to a few hundred bp corresponding to two or less hyper-mCpG tiles referred to as "regular" hyper-mCpG tiles. However, 4% of the hyper-mCpG tiles were linked to larger mCpG perturbation (i.e., two or more hyper-mCpG tiles referred to as 'large' hyper-mCpG tiles) and among such extended signals, local SV or SNV (<1 kb) were significantly more common than expected based on their overall distribution (Supplementary Fig. 10). Specifically, we found that 129 out of 783 (16%) "large" hyper-mCpG tiles had local SVs or SNVs mapped within 1 kb whereas only 96 out of 1057 (9%) "regular" hyper-mCpG tiles had a local SVs or SNVs mapped within the same distance (i.e., 1 kb).

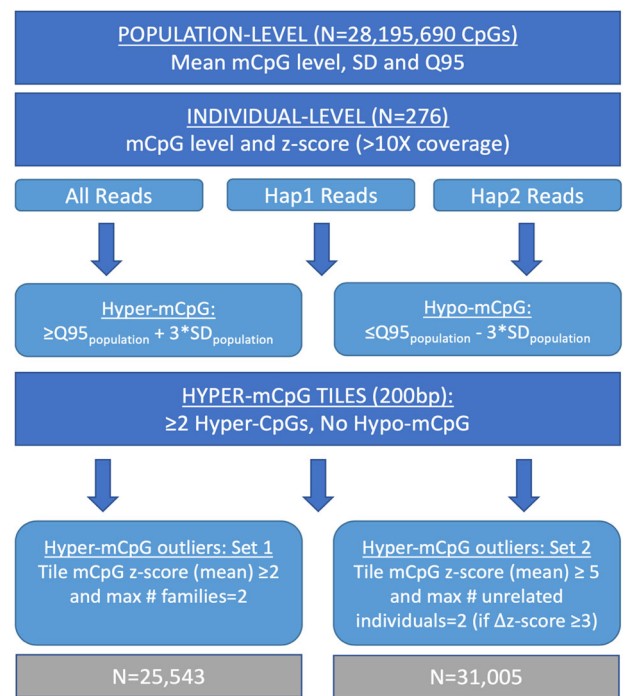

**Fig. 3 | Identification of rare hypermethylation outliers.** Simplified schematic depicting the process to identify rare hypermethylation outliers in our rare disease cohort. SD denotes standard deviation, Q95 denotes the 95th percentile, Hap1 denotes haplotype 1 and Hap2 denotes haplotype 2.

## Rare genetic variation links to hypermethylation in regulatory elements

To assess the extent to which the identified hyper-mCpGs outlier tiles ($N = 56,548$) map to RE we used the largest atlas of regulatory DNA to date that was created from 16 different human tissue types or states ("Methods"). We found that 94% of the extreme hyper-mCpG outliers mapped to a RE compared to 41% for randomly selected non-outlier mCpG tiles ("Methods"), representing a ~twofold enrichment (Fisher's Exact test $P < 2.2E-16$; Fig. 5). We noted largest enrichment for RE (1) shared across 10 or more organ systems (tissue invariant, fold change = 9.7, Fisher's Exact test $P < 2.2E-16$), (2) specific to myeloid/erythroid immune cells (fold-change=5.5, Fisher's Exact test $P < 2.2E-16$) and (3) lymphoid immune cells (fold change = 4.5, Fisher's Exact test $p < 2.2E-16$) in line with the tissue type used here for HiFi-GS (Fig. 5A). As expected, given the high degree of allele-specificity as discussed above, many of these hyper-mCpGs outliers overlapping a RE are linked to rare genetic variation in *cis* mapping either proximal or distal to their potential cognate target genes. For instance, we identified a rare 2.6 kb insertion (INS) (Fig. 5B) in *cis* causing proximal RE hypermethylation at the *DDB2* disease locus (Fig. 5C). Similar effects on RE were observed for SNVs (Supplementary Fig. 11) and SVs (Supplementary Fig. 12) as well as for repeat expansions (Supplementary Figs. 13–15), duplications (Supplementary Fig. 16) and deletions (DELs) (Supplementary Figs. 17 and 18).

## Validation of rare hypermethylation events by full-length cDNA sequencing

Since we focused on rare hyper-mCpG, it is anticipated that the large magnitude of outliers can predominantly be found in the relatively hypomethylated, dynamic[34] REs of the human genome. Consequently, these rare hyper-mCpG outlier may contribute to RE silencing and impact gene expression. To this end, we demonstrated the functional translation of mCpG in a subset of analyses from patients by co-occurrence of allelic rare hyper-mCpG events in

proximal RE and concurrent allelic mRNA silencing. Specifically, we selected hyper-mCpGs outliers (Supplementary Data 4) that mapped within 1 kb of a transcription start site (TSS) and present in an individual where long-read cDNA sequencing was performed on proband-specific induced pluripotent stem cells (iPSCs). The TSS was linked to the closest consensus coding sequence (CCDS) region and reads informative for a heterozygous SNV were selected if covered at more than ten reads for each selected proband and controls. The latter corresponded to all other probands not "carrying" the hyper-mCpG tile. In total, 22 unique hyper-mCpG outliers-transcript pairs across eight probands were informative. We found that in total of 15/22 (68%) of the cases were associated with a differential abundance of the allelic copies of the tested transcript beyond a 45:55 ratio compared to cases where only 3/22 (14%) showed deviations beyond similar ratio (Supplementary Table 1 and Supplementary Fig. 19). Notably, this was achieved in non-blood cells (iPSCs) from the same patients ("Methods"), indicating that some variation exhibits relative to tissue independence.

## Rare hypermethylation outlier facilitates the diagnosis of unsolved rare disease cases

Having genetically and functionally characterized rare hyper-mCpGs tiles, we next queried their relationship with rare disease genes (OMIM) to explore candidate functional changes for unsolved disease. Among the patients, we observed a total of 5438 hyper-mCpG tiles across 4400 OMIM genes (~30 per patient). We anticipate that these outliers can generate noncoding disease variant candidates. Among the OMIM gene overlapping hyper-mCpG tiles, two of the top five highest z-scores mapped to *GNAO1* with a rare intronic SNV (C-to-A MAF < 0.003) associated with the hypermethylation footprint (~500 bp). The hyper-mCpG tiles map to a vertebrate-constrained sequence and ENCODE predicted distal regulatory (enhancer) active across multiple tissue types (Fig. 6A). By accessing HiFi-GS from parents, we noted that both the proband and the father share the SNV with similar hypermethylation effect of the RE. De novo mutations in *GNAO1*, both gain of function and loss of function, cause neurodevelopmental disorder including developmental and epileptic encephalopathy. In this case, the proband (at 2 years) suffers from dysphagia and failure-to-thrive, with paternal family history of seizures. Considering the wide spectrum of *GNAO1*-associated neurodevelopmental disease and the undiagnosed proband's presentation and family history, the variant is a candidate for further study. However, with the expression pattern of *GNAO1* being almost exclusively brain-specific (Supplementary Fig. 20), follow-up analysis requires tissue-targeted experimentations in affected individuals.

Next, we selected hyper-mCpG tiles close to the proximal promoters of OMIM genes (+/− 1 kb) which yielded 1341 regions. We then further restricted to hyper-mCpG tiles with local *cis*-rare SNV or SV for a new subset of 66 regions and noted that most of these genetically linked hyper-mCpG events in our dataset occurred in autosomal recessive OMIM gene promoters. We exemplified this at the *MTHFR* locus showing a paternally inherited rare 5′UTR SNV associated with several hundred base pair of hyper-mCpG (Fig. 6B) and indicating carriership of a potentially deleterious allele. Another set of four hyper-mCpG events (in three patients) near OMIM genes were flagged for follow-up: one tile in *NSD1*, one in *SET* and two adjacent hyper-mCpGs tiles in *DIP2B*. The hyper-mCpG events at the 5′ proximal promoters of the dominant disease genes *NSD1* and *SET* were identified in two patients whose clinical features were not compatible with the syndromes linked to loss of function in these loci. However, further analysis revealed that the hyper-mCpG tiles of the proximal regulatory region of *DIP2B* was immediately adjacent to a previously missed and undiagnosed[3] repeat expansion in a proband with global developmental delay (Fig. 7A). Here, the phasing power and mCpG "staining" ability by HiFi-GS (Fig. 7B) allowed for prioritization and further clinical

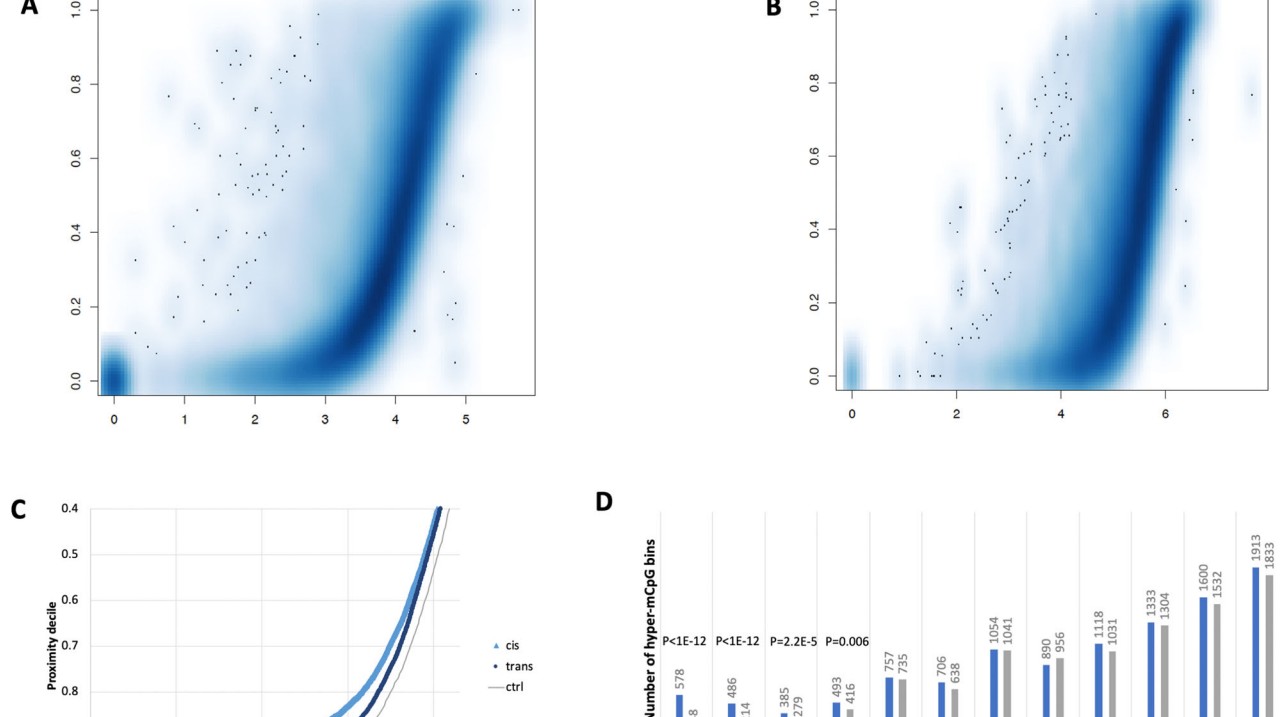

**Fig. 4 | Association of genetic variants with rare hypermethylation outliers.**
Scatter plot depicting the distance (*x* axis, log10 base pair (bp)) between rare hyper-mCpG tiles and closest rare **A** single nucleotide variant (SNV) (with minor allele frequency (MAF) < 0.5% based on gnomAD) and **B** structural variant (SV) (with MAF < 1% based on present study population data). Same process was repeated for, in parallel, for same tile in all other individuals to obtain a percentile rank of tile-variant distances in the population. Empirical *q* value (*y* axis) threshold (dashed black line, *q* < 0.1) corresponding to estimated distance of associated rare variant of rare hyper-mCpG tiles (red solid line). **C** Scatter plot depicting haplotype-resolved distance (*x* axis, bp) between rare hyper-mCpG tiles and closest rare SNV (0.5% MAF gnomAD) in cis (same read, light blue triangle) and trans (opposite read, dark blue round). Same process was repeated for, in parallel, for same tile in all other individuals to obtain a percentile rank (*y* axis) of tile-variant distances in the population. Gray line depicts null distance distribution of rare SNVs (<0.5% MAF gnomAD) in our sample set. **D** Bar graph depicting number of extreme hyper-mCpG tiles (*y* axis) linked to rare SNV in *cis* (same read, blue bars) or *trans* (opposite read, gray bars) based on distance. *P* values from a two-sided binomial test assuming equal number of tiles in *cis* and *trans* for at each distance (*x* axis, bp).

review where the expansion was determined to be within the pathogenetic range (-280–300 repeats, >250 considered pathogenic). This diagnostic finding was clinically validated by triplet-repeat PCR (Methods) and reported to the provider. Within our large HiFi-GS cohort, we had additional unsolved patient specimens with expanded *DIP2B* repeats (Fig. 7B), however all other expansions were below the pathogenic range and remained hypo-mCpG—exemplifying the augmented power of long repeat resolving reads coupled with mCpG detection.

## Discussion

Rare genetic diseases remain a translational challenge in the era of advanced molecular diagnostics. Better data sharing and increasing understanding of expressivity through larger molecularly classified rare disease cohorts have the capacity to discover disease genes and assign function to variants of unknown significance (VUS). However, structural genetic variation and noncoding variation with disease-causing potential are areas of potential unrealized diagnostic yield due to technological and analytical hurdles. Suggestions for closing technology gaps include coupling next-generation sequencing (NGS) with additional assays gathering either transcriptomic[35] or epigenomic[36] data. Integrated genome-wide, high-resolution assessment of any two data modalities coupled with high haploid distinction has not been attempted to date. In our rare disease cohort, we demonstrate several key aspects of combined mCpG and HiFi-GS for the exploration of

disease variation. Known disease-linked and parent-of-origin mCpG is recovered alongside with full GS. Rare variation in the epigenome can be due to multiple sources, including environmental or genetic causes. Here, we show that rare methylation outliers are heritable and can be physically linked to rare noncoding genetic variation in long reads. A substantial fraction of outlier signatures is caused by complex, local rare SV which is challenging to detect by NGS.

We demonstrate the predicted impact of RE activity represented by hyper-mCpG outliers propagating to transcriptional silencing in same patient but in distinct lineage cells, showing that blood-based 5-base HiFi-GS can have multi-tissue relevance. Employing an analytical approach to identifying population variation, we can limit the search to reasonable sets of candidate alleles for prioritization of manual curation where our stringent filter already identified previously missed diagnoses for a rare disease involving *DIP2B*. However, the potential discoveries by this platform, even within this dataset with follow-up validation, will expand to additional disease alleles and demonstrates the ability to link noncoding variation to clinical evaluation in rare diseases.

## Methods
### Ethical declarations
The study complies with all relevant ethical regulations as approved by the Children's Mercy Institutional Review Board (IRB) (Study # 11120514). Informed written consent was obtained from all participants

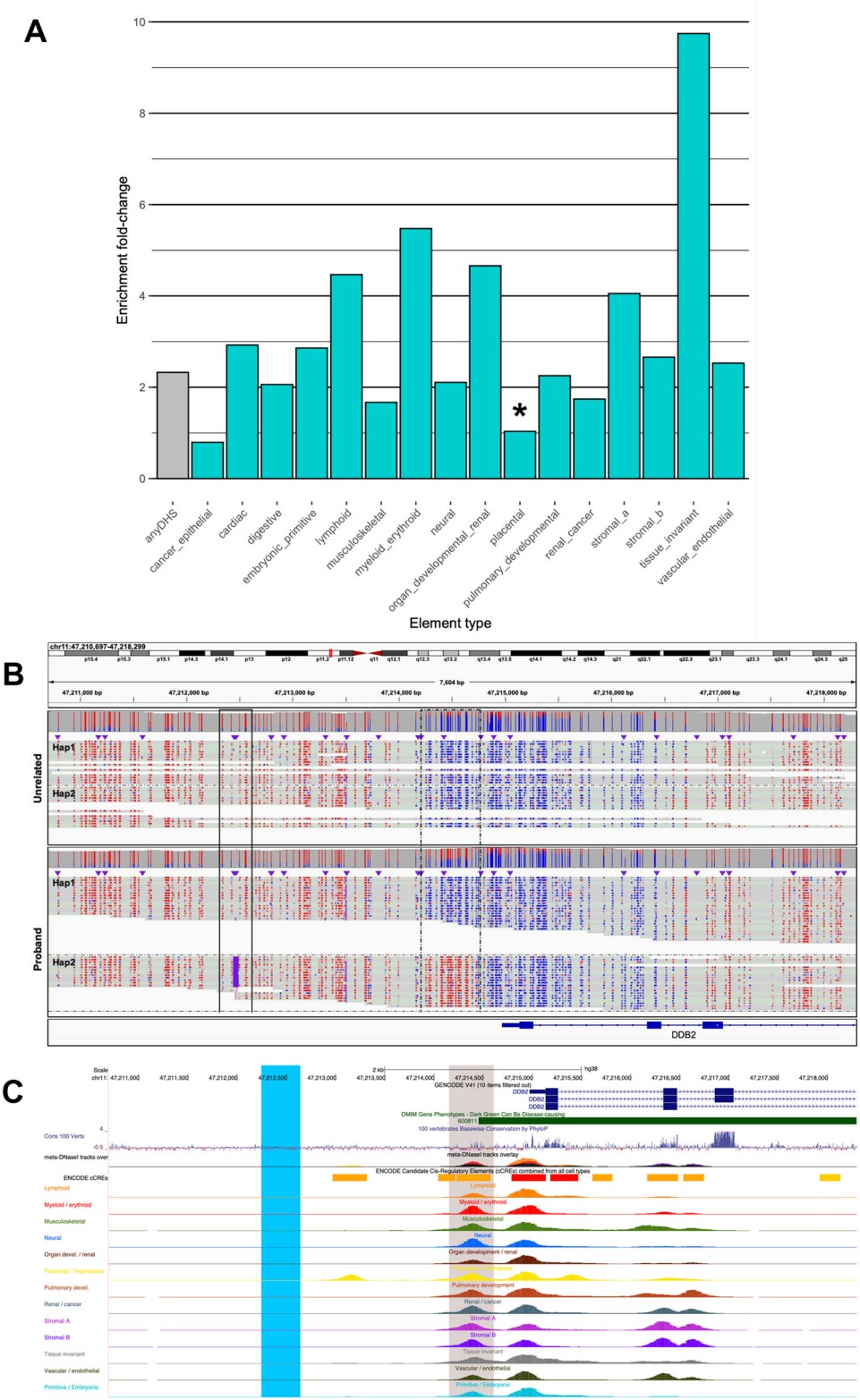

**Fig. 5 | Regulatory element annotation of rare hypermethylation outliers. A** Bar graph depicting the fold enrichment (*y* axis) of the overlap of regulatory elements (assessed by DNase I hypersensitive site (DHS) mapping) with extreme hyper-mCpG tiles compared to background control tiles. Enrichment is presented as either overlapping any DHS site (gray bar) or specific to tissue states (turquoise bars). Significance is based on two-sided Fisher's exact test where all element types have *P* values < 2.2E-16, except "cancer_epithelial" (*P* = 2.00e-08) and placental (*P* = 0.251, not significant, indicated with asterisk). **B** Genomics view of 7.6 kb at the *DDB2* disease locus comprising an insertion (black box, solid like) resolved by HiFi-genome sequencing (GS) that results in 1 kb proximal promoter hypermethylation (black box, dashed line). Haplotype-resolved CpG modification coloring (blue indicating unmethylated CpG prediction and red indicating methylated CpG prediction with color intensity corresponding to base modification probabilities) is shown in proband and unrelated sample. Hap1 denotes haplotype 1 and Hap2 denotes haplotype 2. **C** Zoomed in genomics view depicting regulatory element (DHS) annotation of the hyper-mCpG tile (gray box) nearby causal insertion (blue box).

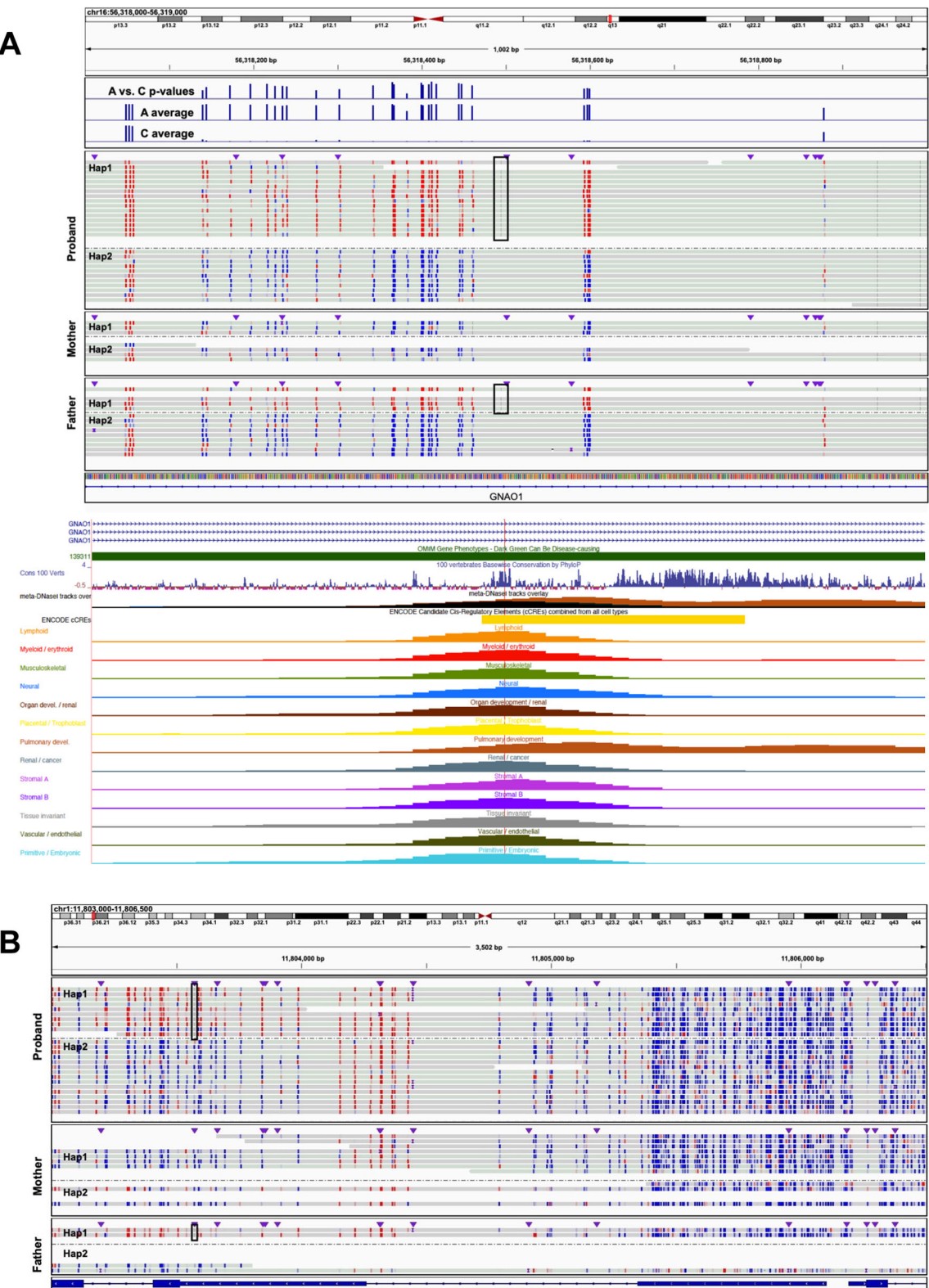

prior to study inclusion and included consenting for collecting biospecimens for the purpose of deriving patient-specific cell lines. Participants were not compensated for study participation.

## Study cohort

The study cohort described includes 1243 affected probands from 1078 families, with a total of 1367 individuals (detailed in Supplemental Data 1) enrolled in the Genomic Answers for Kids program[3]. Probands age at enrollment ranged from 0 to 32 years (median 6 years) with 47% being female and 53% male, respectively. The sex of participants was obtained based on self-report and subsequently confirmed by sequencing. No sex-specific analysis was performed as this was out of the scope of the study. We focus all analysis on autosomes thus the findings are applicable to both sexes. Patients with a suspected

**Fig. 6 | Disease genes with rare hypermethylation event in regulatory elements.**
**A** Genomic view of ~1 kb on chromosome 16 at the *GNAO1* locus showing a paternally inherited rare (minor allele frequency <0.3%) intronic C-to-A single nucleotide variant (SNV) at chr16:56,318,495 (black rectangle). Haplotype-resolved CpG modification coloring (blue indicating unmethylated CpG prediction and red indicating methylated CpG prediction with color intensity corresponding to base modification probabilities) is shown in the complete trio. Hap1 denotes haplotype 1 and Hap2 denotes haplotype 2 in the respective sample. Top track shows *P* values (*y* axis, −log10(0–31)) from two-sided Fisher's exact test for CpG methylation differences in rare A versus common C-allele carrying reads, respectively. Second and third track from the top shows the average CpG methylation rate (*y* axis, 0–100%)

across all A (rare) and C (population prevalent) carrying reads, respectively. Genome browser view depicts the rare SNV (red line) overlapping a vertebrate-constrained sequence and conserved regulatory element active across tissue types. **B** Genomic view of ~3.5 kb on chromosome 1 at the *MTHFR* locus showing a paternally inherited rare 5'UTR SNV (black box) which maps into an extended ~1 kb extreme hyper-mCpG tile at the *MTHFR* proximal promoter. Haplotype-resolved CpG modification coloring (blue indicating unmethylated CpG prediction and red indicating methylated CpG prediction with color intensity corresponding to base modification probabilities) is shown in the complete trio. Hap1 denotes haplotype 1, and Hap2 denotes haplotype 2 in the respective sample.

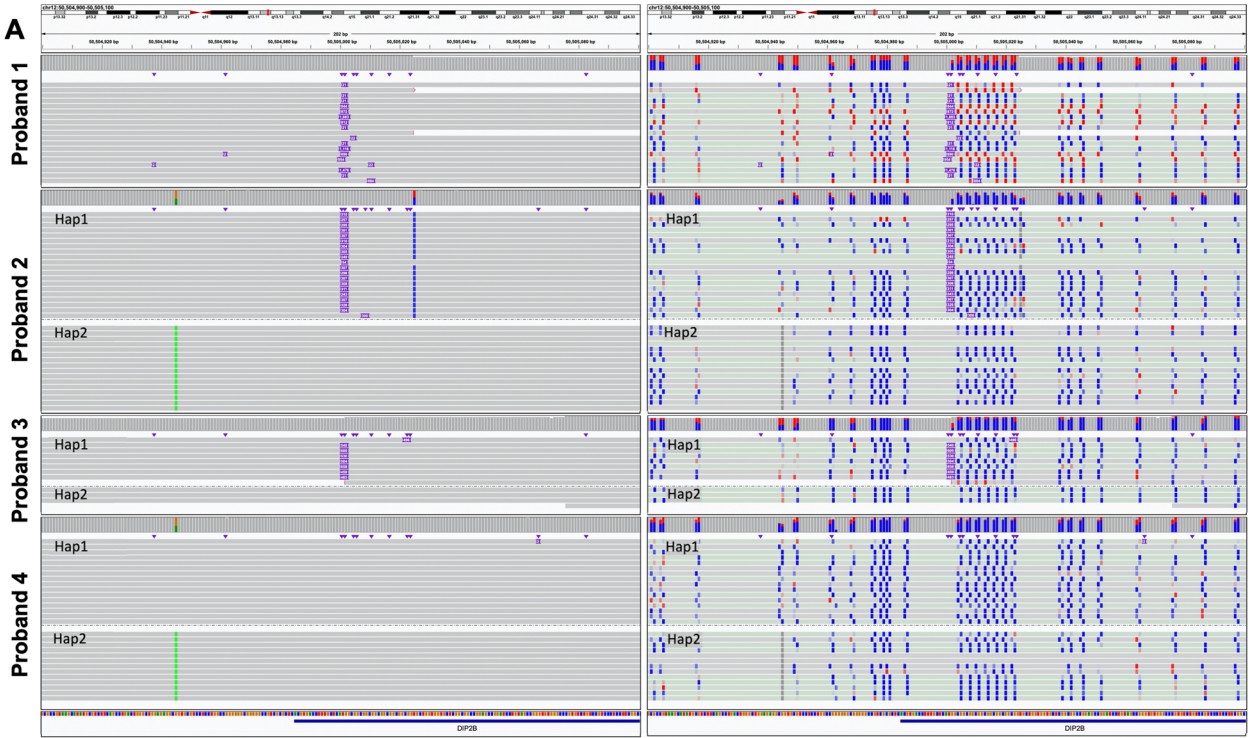

**Fig. 7 | Rare hypermethylation event in a previously undiagnosed repeat expansion. A** HiFi-genome sequencing (GS) reads across four individuals detecting (1) large (Proband 1), (2 and 3) expanded (Haplotype-resolved, Proband 2 and Proband 3) and (4) no repeats (Haplotype-resolved, Proband 4) at the *DIP2B* locus.

Hap1 denotes haplotype 1 and Hap2 denotes haplotype 2 in respective sample. **B** CpG modification coloring of HiFi-GS reads (blue indicating unmethylated CpG prediction and red indicating methylated CpG prediction with color intensity corresponding to base modification probabilities) across the four individuals.

rare disease were referred from multiple different specialties, with the largest proportion nominated by Clinical Genetics, followed by Neurology. A continuum of pediatric conditions is represented, ranging from congenital anomalies to more subtle neurological and neurobehavioral clinical presentations later in childhood. Of the 1243 affected patients, 141 had a known genetic diagnosis at the initiation of the study.

## Sample collection and preparation
Whole blood was obtained from each study participant in EDTA and sodium heparin collection tubes for DNA and peripheral blood mononuclear cell isolation (PBMC), respectively. DNA was isolated using the chemagic™ 360 automated platform (PerkinElmer) and stored in −80 °C. PBMCs were isolated using a RoboSep-S (StemCell) and the EasySep Direct Human PBMC Isolation Kit (StemCell 19654RF). After automated separation, the enriched cell fraction was centrifuged at $300 \times g$ for 8 min then resuspended in 1 mL of ACK Lysing Buffer (ThermoFisher A1049201) and incubated at room temperature for 5 min. The cell suspension was diluted with 13 mL of PBS (Thermo-Fisher 10010023) supplemented with 2% FBS (Cytiva SH30088.03HI)

then centrifuged at $300 \times g$ for 8 min. The cell pellet was resuspended in 1 mL of PBS + 2% FBS, and cell count and viability were assessed using a Countess II automated cell counter (ThermoFisher). Cells were centrifuged at $300 \times g$ for 8 min and resuspended in 1 mL of cold CryoStor CS10 (StemCell 07930) then transferred to a cryogenic storage vial. Cells were frozen slowly in a Corning CoolCell FTS30 placed at −80 °C overnight then transferred to liquid nitrogen vapor the following day for long-term storage. Human patient-specific induced pluripotent stem cells (iPSCs) were generated from a subset of the patient's PBMCs using episomal vectors. In brief, PBMCs were maintained in StemSpan SFEM II (STEMCELL Tech, 09605) media plus 1× Antibiotic–Antimycotic (ThermoFisher, 15240062) for 3–5 days, followed by nucleofection of the following episomal plasmids: pCXLE-hOCT3/4-shp53-F (Addgene, 27077), pCXLE-hSK (Addgene, 27078), pCXWB-EBNA1 (Addgene, 37624) and pCXLE-hUL (Addgene, 27080). After nucleofection, transfected PBMCs were plated onto 35-mm matrigel (Corning, 354277) coated tissue culture dishes in Stemspan SFEM II media supplemented with 10 µM Y-27632 (Tocris, 1254). Two days later, ReproTESR (STEMCELL Tech, 05926) was added, and plates were centrifuged at $50 \times g$ for 30 min. Every other day fresh ReproTESR

was added until iPSC colonies were visible. Once colonies were visible ReLeSR (STEMCELL Tech, 05872) was used to isolate iPSC colonies. iPSC colonies were maintained in mTeSR1 complete media (STEMCELL Tech, 85850) on matrigel coated tissue culture plates and refed every other day or as needed until ready. Cells were cultured at 37 °C in 5% $CO_2$.

## PacBio HiFi long-read genome sequencing and analysis

In total of ~5 ug of DNA per sample was sheared to a target size of 14 kb using the Diagenode Megaruptor3 (Diagenode, Liege, Belgium). SMRTbell libraries were prepared with the SMRTbell Express Template Prep Kit 2.0 (100-938-900, PacificBiosciences, Menlo Park, CA) following the manufacturer's standard protocol (101-693-800) with some modifications as follows[3]: Ligation is completed overnight (20 °C hold overnight, 65 °C 10 minutes with a 4 °C hold) and nuclease v1 is used, with an incubation of 37 °C for 1 h. Fragments longer than 10 kb were selected using the Sage Science PippinHT (Sage Science, Beverly, MA). Libraries were sequenced on the Sequel IIe Systems using the Sequel II Binding Kit 2.0 (101-842-900) or 2.2 (102-089-000) and Sequel II Sequencing Kit 2.0 (101-820-200) with 30 hr movies/SMRT cell. Samples were sequenced to a target of >10X coverage.

Circular consensus reads were generated with ccs v6.3 (https://github.com/PacificBiosciences/ccs) using the "--hifi-kinetics" option to generate consensus kinetics tags, and primrose v1.1 (https://github.com/PacificBiosciences/primrose) was used to predict 5mC modification of each CpG motif and generate base modification ("MM") and base modification probability ("ML") BAM tags. HiFi read mapping, variant calling, and genome assembly were performed using a Snakemake workflow (https://github.com/PacificBiosciences/pb-human-wgs-workflow-snakemake). HiFi reads were mapped to GRCh38 (GCA_000001405.15) with pbmm2 v1.9 (https://github.com/PacificBiosciences/pbmm2). Structural variants were called with pbsv v2.8 (https://github.com/PacificBiosciences/pbsv) with "--hifi --tandem-repeats human_GRCh38_no_alt_analysis_set.trf.bed" options to pbsv discover and "--hifi -m 20" options to pbsv call. Small variants were called with DeepVariant v1.3 following DeepVariant best practices for PacBio reads (https://github.com/google/deepvariant/blob/r1.3/docs/deepvariant-pacbio-model-case-study.md) and locally phased with WhatsHap v1.0[37]. Local phase haplotype tags ("HP") were added to the aligned BAM by WhatsHap v1.0. Pileup-based consensus methylation sites and probabilities were generated by the script "aligned_bam_to_cpg_scores.py" from pb-CpG-tools v1.1.0 (https://github.com/PacificBiosciences/pb-CpG-tools/) with the "-q 1 -m denovo -p model -c 10" options. Reads were visualized in IGV version 2.15.2.

## Whole-genome bisulfite sequencing and analysis

The same source of DNA used for PacBio HiFi long-read genome sequencing was used for whole-genome bisulfite sequencing. Whole-genome sequencing libraries were generated from 1ug of genomic DNA spiked with 0.1% (w/w) unmethylated λ DNA (Promega) previously fragmented to 300–400 bp peak sizes using the Covaris focused-ultrasonicator E210. Fragment size was controlled on a Bioanalyzer DNA 1000 Chip (Agilent) and the KAPA High Throughput Library Preparation Kit (KAPA Biosystems) was applied. End repair of the generated dsDNA with 3′- or 5′-overhangs, adenylation of 3′-ends, adaptor ligation and clean-up steps were carried out as per KAPA Biosystems' recommendations. The cleaned-up ligation product was then analyzed on a Bioanalyzer High Sensitivity DNA Chip (Agilent) and quantified by PicoGreen (Life Technologies). Samples were then bisulfite converted using the EZ DNA Methylation–Lightning Gold Kit (Zymo), according to the manufacturer's protocol. Bisulfite-converted DNA was quantified using OliGreen (Life Technologies) and, based on quantity, amplified by 9–12 cycles of PCR using the Kapa Hifi Uracil+DNA polymerase (KAPA Biosystems), according to

the manufacturer's protocol. The amplified libraries were purified using Ampure Beads and validated on Bioanalyzer High Sensitivity DNA Chips and quantified by PicoGreen. Libraries were sequenced on the Illumina NovaSeq6000 system using 150-bp paired-end sequencing.

Whole-genome bisulfite sequence reads were pre-processed with fastp to trim adapter and low-quality bases, then alignment was performed with Illumina's DRAGEN aligner followed by post-processing with samtools 1.9, Picard 2.17.8, Bismark_v0.20.0[38] and Bis-SNP 0.82.2[39] for marking of duplicates, methylation calling and SNV calling. To avoid potential biases in downstream analyses, we applied our benchmark filtering criteria as follows; ≥5 total reads, no overlap with SNPs (dbSNP 137), ≤20% methylation difference between strands, no overlap with DAC Blacklisted Regions (DBRs) or Duke Excluded Regions (DERs) generated by the ENCODE project: (http://hgwdev.cse.ucsc.edu/cgi-bin/hgFileUi?db=hg19&g=wgEncodeMapability). Methylation values at each site were calculated as total (forward and reverse) non-converted C-reads over total (forward and reverse) reads. CpGs were counted once per location combining both strands together.

## Correlation of CpG methylation across platforms

For sample-wise correlation ($N = 93$), CpGs sequenced at >20× coverage across both platforms were selected and used to measure the linear correlation using Pearson correlation coefficient ($r$).

For CpG-based correlations, the HiFi-GS dataset was used to select the 500 CpGs with the highest standard deviation (SD) for the methylation rate. From the set of 93 samples with both HiFi-GS and WGBS, the methylation rates at these 500 CpGs were extracted. Then, for each CpG, the correlation between the HiFi-GS methylation rate and the WGBS methylation rate was computed using Pearson correlation coefficient ($r$). Also, for each CpG, a control correlation rate was computed by shuffling the methylation rates of all 93 samples and then computing the correlation between the HiFi-GS methylation rate and the (now-shuffled) WGBS methylation rate, whenever a matching methylation rate was available.

## Classification of rare extreme hypermethylation outliers

The HiFi-GS dataset was used to calculate mean, standard deviation (SD) and 95th quantile (Q95) for each of the measured CpGs (Supplementary Data 2). Then, for each HiFi-GS sample, the methylation rate and z-score for each CpG at 10× coverage was computed using all HiFi reads sequenced. This was repeated for each sample for each CpG at 10× read coverage using only reads phased to haplotype 1 (hap1) and repeated a third time using only the reads phased to haplotype 2 (hap2). Consequently, methylation rate and z-score were obtained for each combination of individual, CpG chromosomal position and phasing mode (all, hap1 or hap2) whenever the corresponding read depth was 10 or greater. A CpG position for an individual and phasing mode (all, hap1 or hap2) was flagged as hypermethylated if the methylation rate deviated three SD higher than the Q95 for that CpG. A CpG position for and individual and phasing mode was flagged as hypomethylated if deviating three SD lower than the Q5 for that CpG. Then, all CpGs for an individual and phasing mode were grouped into 200 bp tiles across all autosomes. A 200 bp tile for an individual and phasing mode was identified as an extreme hyper-mCpG tile if it contained two or more hypermethylated CpGs and no hypomethylated CpGs. Extreme hyper-mCpG tiles were further reported as rare in two ways: (1) hyper-mCpG tiles at the same position from any phasing mode are present in two or fewer unrelated individuals, reporting the hyper-mCpG tiles for an individual when the average methylation rate z-score is greater than 2 across all CpGs for the hyper-mCpG tile of that individual for a phasing mode; or (2) hyper-mCpG tiles at a position are potentially present in three or more unrelated individuals, but only hyper-mCpG tiles with an average methylation rate z-score greater than 5 across the CpGs in the hyper-mCpG tile of that individual for a

phasing mode was kept. If the hyper-mCpG tile was present across multiple unrelated individuals, a minimum separation of three z-score units across the unrelated samples was required allowing only a maximum of two samples per tile.

The hyper-mCpG tiles were annotated with the closest OMIM gene position, the closest gene transcription start size (TSS) and overlapping DNase I hypersensitivity sites (DHS)[40]. For enrichment analysis of extreme hyper-mCpG tiles in regulatory element annotated by DHS, two sets of randomly selected hyper-mCpG tiles were created requiring the tile being present in at least 5 samples (out of 276) and with a mean methylation rate z-score per tile corresponding $-1 <= z <= 1$ and $-2 <= z <= 2$, respectively. These criteria resulted in a total of 10,129,897 and 10,134,018 hyper-mCpG tiles, respectively, of which 50,000 were randomly selected from each set and used as background control.

## Mapping genetic variation to extreme hypermethylation outliers

The distance from the extreme bin to the closest gnomAD[41] high-confidence variant in the sample with the extreme bin was compared to the distance from the same bin coordinates to the closest gnomAD high-confidence variant in all other samples (with a variant on that chromosome) to obtain a percentile rank. The distance to the closest phased (via WhatsHap) heterozygote gnomAD high-confidence variant on hap1 in the sample was also compared with the extreme bin against the distance from that bin coordinate to the closest phased hap1 gnomAD variant in all other samples to obtain a percentile rank, and similarly for the closest phased heterozygote variant on hap2.

The excess familial sharing of hypermethylation outliers was observed to be significant in random permutations among samples that had family members available. Specifically, among called hypermethylation outliers (max two called per window) there were total 2465 observations in samples with related individual among to cohort, of these 290 were shared with the related family member (11.8%). The familial relationships (family labels) were then permuted in the hypermethylation tiles and calculated random overlap of familial sharing observing on average 94.2 (range 36–58 per permutation run) such overlaps by chance indicating an 3.079x excess of familial sharing (290/94.2).

## PacBio HiFi long-read transcript (isoform) sequencing and analysis

RNA was isolated from 17 iPSC samples using a RNeasy Mini Kit (Qiagen, Cat. No. 74104), following the Quick Start Protocol associated with the kit. The RNA concentration for each sample was determined with a Qubit RNA BR Assay Kit (ThermoFisher, Q10210) and the RIN was determined with a RNA ScreenTape (Agilent, Cat. No. 5067–5577 and 5067–5576) on the TapeStation platform. A maximum of 300 ng of RNA with a RIN score greater than 7.0 was aliquoted from each sample to be used as an input into cDNA synthesis for Iso-Seq library preparation. If the concentration of RNA was too low for there to be 300 ng of RNA in 7 μL, 300 ng of RNA was aliquoted into a 1.5-mL tube and then the sample was concentrated using a vacuum concentration system without heating until the sample was less than or equal to 7 μL in volume. The RNA underwent cDNA synthesis with a NEBNext Single Cell/Low Input cDNA Synthesis & Amplification Module (NEB, Cat. No. E6421S) and an Iso-Seq Express Oligo Kit (PacBio, Cat. No. 101-737-500) as described in the Iso-Seq Express Template Preparation for Sequel and Sequel II Systems protocol from PacBio. The samples were not amplified with barcoded primers during the cDNA amplification steps since the downstream libraries would not be multiplexed for sequencing. The amplified cDNA was purified for long transcripts greater than 3 kb in length after the first cDNA amplification step. The quantity of cDNA was determined with a Qubit dsDNA HS Assay Kit (ThermoFisher, Cat. No. Q32854) and if the cDNA yield was less than 160 ng, the required input for the Sequel II system, the samples were reamplified following the procedure described in "Appendix 1" of the Iso-Seq protocol from PacBio. NEBNext High-Fidelity 2X PCR Master Mix (NEB, Cat. No. M0541S) used in the PCR reamplification master mix and after reamplification, the cDNA was bead-cleaned following the low-yielded sample procedure. The cDNA was quantified again to ensure that there was adequate yield for SMRTbell library preparation. In total, 160–500 ng of cDNA from each sample was used as an input into the SMRTbell library preparation protocol without pooling the cDNA. The final concentration of the libraries was determined with a Qubit dsDNA HS Assay Kit and the library size was determined with a High Sensitivity D5000 ScreenTape (Agilent, Cat. No. 5067–5592 and 5067–5593) on the TapeStation platform. Libraries were sequenced on the Sequel IIe Systems using the Sequel II Binding Kit 2.0 (101-842-900) or 2.1 (101-820-500) and Sequel II Sequencing Kit 2.0 (101-820-200) with 30 h movies/SMRT cell at 90pM loading with 1 cell/sample. The IsoSeq3 pipeline (https://github.com/PacificBiosciences/IsoSeq) was used to generate full-length non-concatemer (FLNC) reads which were then aligned to the reference genome (GRCh38) using minimap2 with the argument -ax splice:hq. Reads overlapping a CCDS region and informative of a heterozygous SNV were kept and each allele was count for each read covered by at least 5×.

## Clinical validation

"Short" PCR amplification of the normal allele of the FRA12A/*DIP2B* repeat–containing region was performed with the aid of 2X Failsafe buffer J (received with FailSafe enzyme) with use of primers derived from the sequences flanking the repeat (DIP2B-CGG-F primer, 5'-GTC TTC[1]AGCCTGACTGGGCTGG-3', and DIP2B-CGG-R, primer 5'-CCG G[1]CGACGGCTCCAGGCCTCG-3'). 95 °C–3'; (95 °C–1'; 60 °C–1'–0.5/ cycle; 72 °C–1') × 10; (95 °C–1'; 55 °C–1'; 72–1') × 25; 72 °C–1', 4 °C ∞). The PCR products were electrophoresed on a 1.5% agarose gel. Triplet-repeat PCR was also performed by primed PCR according to the principles described previously[42,43] to detect CAG-repeat expansion.

Three different primers were added to the PCR mixture: a single forward fluorescently labeled primer (DIP2B-CGG-F) and a combination of two reverse primers (P4, 5'-TACGCATCCCAGTTTGAGACGGCC GCCGCCGCCGCCGC-3', and P3, 5' -TACGCATCCCAGTTTGAGACG-3') in a 1:10 ratio. The reverse primer P4 anneals at different sites of the CGG repeat, which produces PCR products of different lengths that differ from each other by a multiple of three residues. After depletion of the P4 primer, the P3 primer takes over and amplifies the PCR products of different lengths. TP-PCR conditions (95 °C–7' (95 °C–1', 63 °C + 0.5/cycle–1', 72 °C–1') × 10, (95 °C–30", 56 °C–30", 72 °C–1' + 10"/cycle) × 25, 72 °C– 10',4 °C–∞).

Diluted PCR products (1:5) were mixed with 1200LIZ and Hi-Di Formamide, PCR products were size fractionated on a Prism ABI 3500 DNA sequencer (Applied Biosystems).

## Reporting summary

Further information on research design is available in the Nature Portfolio Reporting Summary linked to this article.

## Data availability

The 5-base HiFi-GS, HiFi long-read transcript sequencing (IsoSeq) and WGBS raw and processed data generated in this study have been deposited in the dbGAP (https://www.ncbi.nlm.nih.gov/gap/) database under accession code phs002206.v4.p1 [https://www.ncbi.nlm.nih. gov/projects/gap/cgi-bin/study.cgi?study_id=phs002206.v4.p1]. Raw and processed data are available under restricted access due to IRB regulations and informed consent limiting access to users studying genetic diseases. Data access are provided by dbGAP (https://dbgap. ncbi.nlm.nih.gov/aa/wga.cgi?page=login) for certified investigators with local IRB approval in place. The reference genome GRCh38

(GCA_000001405.15) used in this study is available at ftp://ftp.ncbi. nlm.nih.gov/genomes/all/GCA/000/001/405/GCA_000001405.15_ GRCh38/seqs_for_alignment_pipelines.ucsc_ids/GCA_00001405.15_ GRCh38_no_alt_analysis_set.fna.gz. WGS and HTG sequences that are not part of the human reference genome, GRCh38 (including the ALT sequences) was added using https://www.ncbi.nlm.nih.gov/assembly/ GCA_000786075.2/. The Gnomad v2.1 used in this study is available via Nirvana 3.18.1 (https://github.com/Illumina/Nirvana/).

## Code availability

Only publicly available tools were used in data analysis as described wherever relevant in "Methods".

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

## Acknowledgements

We would like to thank all families for participating in the Genomic Answers for Kids study. This work was made possible by the generous gifts to Children's Mercy Research Institute and Genomic Answers for Kids program at Children's Mercy Kansas City. We also would like to thank Nick Nolte, Dan Louiselle and Rebecca Biswell for their work in sample processing, Laura Puckett and Adam Walters for their work in library preparation and sequencing and the clinical coordination team led by Bradley Belden for their work in clinical coordination. We also would like to thank PacBio for sequencing support for a subset of the samples. T.P. holds the Dee Lyons/Missouri Endowed Chair in Pediatric Genomic Medicine, and E.G. holds the Roberta D. Harding & William F. Bradley, Jr. Endowed Chair in Genomic Research.

## Author contributions

T.C., E.G., and T.P. conceived and designed the study; E.F., A.SA.C., T.N.Z., T.K.T., G.E., I.T., M.G., and W.B.R. prepared, provided and/or analyzed clinical samples and associated data; W.A.C., W.J.R., R.H., D.M.P., C.T.S., B.K., C.B., C.S-S., B.Y., J.J.J., and A.M.W. provided bioinformatics support; J.C.M. performed experiments; W.A.C., A.F.J., S.T.Y., E.G., and T.P. analyzed the data and interpreted results of experiments; W.A.C., A.F.J., E.G., and T.P. prepared figures and drafted the manuscript; E.G., and T.P. edited and revised the manuscript; All authors approved the final version of the manuscript.

## Competing interests

W.J.R., R.J.H., D.M.P., C.T.S., and A.M.W. are current or past employees of PacificBiosciences. The remaining authors declare no competing interests.

## Additional information

[1]Department of Pediatrics, Genomic Medicine Center, Children's Mercy Kansas City, Kansas City, MO, USA. [2]Pacific Biosciences, Menlo Park, CA, USA. [3]Department of Pediatrics, School of Medicine, University of Missouri Kansas City, Kansas City, MO, USA. [4]Department of Pathology and Laboratory Medicine, Children's Mercy Kansas City, Kansas City, MO, USA. [5]Center for Human Genetics and Genomics, Department of Pediatrics, Department of Neuroscience and Physiology, New York University Grossman School of Medicine, New York, NY, USA. [6]Child Health Research Institute, Department of Pediatrics, Nebraska Medical Center, Omaha, NE, USA. [7]Children's Mercy Research Institute, Kansas City, MO, USA. ✉e-mail: egrundberg@cmh.edu; tpastinen@cmh.edu

