## [Peer Review File · Nature Communications]

Direct haplotype-resolved 5-base HiFi sequencing for genome-wide profiling of hypermethylation outliers in a rare disease cohortREVIEWER COMMENTS

Reviewer #1 (Remarks to the Author):

Cheung and colleagues present a study describing how the PacBio long-read HiFi genome sequencing allows simultaneous detection of DNA sequence polymorphisms and DNA methylation over long parentally-separated haplotypes.

They have a phenomenal dataset of 1,184 individuals on whom short read whole genome bisulphite sequencing was performed, 276 individuals on whom the new PacBio sequencing was performed, and 93 with “corresponding measurements” which I interpret to mean both assays. This is one of many frustratingly vague descriptions in the manuscript that require a lot of guesswork on the part of the reader.

The utility of the PacBio sequencing in rare disease diagnostics is amply demonstrated by showing the association of repeat expansions and acquisition of DNA methylation locally.

The central message of the paper talks about ‘hypermethylation outliers’, but this is extraordinarily opaque to the reader. This text really doesn’t convey enough information “In addition, we extracted the top 500 most variable autosomal CpGs based on a subset of the HiFi-GS samples sequenced at high depth (N=139) for CpG-based correlations. Similarly, we noted high correlations compared to randomly permuted values (median Pearson $R=0.86$, Extended Data 1)”, while the Methods section Classification of extreme hypermethylation outliers is likewise very difficult to follow – I have no idea how I would reproduce their analytical approach.

Getting lost at this early stage creates problems for the rest of the manuscript. It’s not clear what is being compared to get the results of Extended Data 1, or whether the permutation approach is appropriate.

Another problem is the way the DNA methylation data are illustrated in the figures. The wiggle tracks of Extended Data 2 are clear, but when we start looking at the IGV screenshots of Extended Data 3-19, the impression is that the red (methylated) sites are not only more strikingly obvious, it looks like the number of CGs with calls is decreased when the site is unmethylated – the data look faded and sparse. As a result, we’re asked to believe that there is a change in DNA methylation when the figures look more like a failure to assign a methylation state.

When these graphical outputs are compared with Figure 4 of Ni et al. (<https://www.biorxiv.org/content/10.1101/2022.02.26.482074v1.full>), the difference is striking. This discrepancy is unexplained and really limits the confidence in the results obtained.

Figure 1 is likewise much too difficult to understand. Despite spending a significant amount of time trying to make sense of it I remain unable to explain what is being conveyed.

Figure 2 sounds like it is showing something valuable, a candidate regulatory variant at genes causing rare disease, but panel (a) suffers from unlabelled wiggle tracks and the strange faded depiction of what is described as unmethylated DNA, while panel (b) unfortunately failed to expand the gene annotation track, making it difficult to appreciate that the event is actually promoter-proximal. The complete lack of labelling for panel (c) is a major problem.

Typically a manuscript as poorly presented as this should be recommended for rejection, it comes across as an unsupervised draft. What gives me pause is the recognition that the underlying data are probably extremely interesting, with associations between genomic variability and local DNA methylation. I would encourage the authors to start again, completely rewrite the manuscript, generate professional figures, explicit methods and descriptive text, and put that version in front of the reviewers for a re-appraisal.

Reviewer #2 (Remarks to the Author):

Review of Direct haplotype-resolved 5-base HiFi sequencing for genome-wide profiling of hypermethylation outliers in a rare disease cohort by Cheung and colleagues. The authors present work using PacBio HiFi sequencing to create haplotype-resolved data from a cohort of 276 samples / 152 families with rare diseases to identify rare hyper methylation events. A large number of events were allele-specific, and the authors demonstrated heritability. The authors identified an average of 30-40 hyper cpGs per individual that overlapped rare disease genes—more than I would have expected, but perhaps not surprising given the definition of a tile they used and the relaxed criteria for a hyper cpG. In general, I find the paper interesting and an important contribution to the field but with results that will need to be investigated further and validated by other studies. The methods allow for understanding of how the authors did the work for those familiar with long-read sequencing tools, but likely will be difficult for others to reproduce. I would encourage the editor to ask for additional data to be released to make the study replicable (see below). Besides data availability, I have no major concerns, just minor comments.

The authors relaxed the definition of rare hyper cpg site in order to increase the number of candidate loci from 25,543 to an additional 31,726 sites. If they had not done this, would this have affected any of the major findings from the results? For example, would the DIP2B result have been flagged?

How many of these hyper-cpg tiles were within annotated transposable elements? Is there a significant enrichment for LINEs? I've seen increases in local methylation, similar to the tiles the authors use, in these but have never managed to convince myself it correlates with anything meaningful.

Was the number of hyper cpg sites higher, lower, or similar in the 141 individuals with a known genetic diagnosis?

I may just be missing this, but how did the authors handle repetitive regions where short reads align poorly? Were these excluded? Segmental duplications, for example.

Did the authors consider controlling for age when looking for hyper-cpg sites? We see age-related differences in our data, so I suspect it is in yours as well.

In all figures it might be difficult for readers not familiar with IGV and the phased view to discern the haplotypes. I would encourage the authors to add some kind of label to the haplotypes that makes them more clear beyond the small 1 and 2 from IGV.

I found the IGV screenshot in figure 3C difficult to read as there is a lot of data—specifically, I can't see how big those inserts are.

In the spirit of data sharing and also making life easier for everyone I would encourage the authors to provide the tile windows/coordinates along with the average methylation for that tile in this population. Others could use this data as a comparator (myself included) for our own studies.

AUTHOR RESPONSE OVERVIEW

We thank both reviewers for their careful review and constructive comments and are pleased to see the mutual enthusiasm about our study. We acknowledge their requests and concerns and have thoroughly and extensively responded to all the overarching concerns. Additional specific comments from each Reviewer are similarly addressed and outlined further below. The overarching concerns raised by the reviewers included:

Overall Presentation

We have completely reformatted the manuscript and added more detailed descriptions of all methods used and analyses performed. Specifically, we have updated all figures to visualize the methylation signatures more clearly using an updated version of the IGV software.

Data Sharing

As indicated in the submitted version, all data are deposited to dbGAP under the accession number phs002206.v4.p1. In addition, as requested by Reviewer#2, we are now not only providing the tile coordinates but also average methylation rates for combined reads as well as separated for each haplotype for each hyper-mCpG tile. These are presented in **Supplementary Table 4; Supplementary Table 6 and Supplementary Table 8.**

REVIEWER COMMENTS

Reviewer #1 (Remarks to the Author):

“Cheung and colleagues present a study describing how the PacBio long-read HiFi genome sequencing allows simultaneous detection of DNA sequence polymorphisms and DNA methylation over long parentally-separated haplotypes.”

“They have a phenomenal dataset of 1,184 individuals on whom short read whole genome bisulphite sequencing was performed, 276 individuals on whom the new PacBio sequencing was performed, and 93 with “corresponding measurements” which I interpret to mean both assays. This is one of many frustratingly vague descriptions in the manuscript that require a lot of guesswork on the part of the reader.”

- We apologize for unclear description of our sequencing dataset. We are now more clearly stating that we generated a comprehensive mCpG dataset from 1367 enrolled participants utilizing WGBS (N=1184) and HiFi-GS (N=276). Specifically, 1091 participants were profiled by WGBS only, 183 by HiFi-GS only and 93 by both platforms, respectively. We present this in the Results section, under the newly added paragraph **“Benchmarking HiFi-GS for parallel genome and methylome assessments in a rare disease cohort” on Page 5.**

“The utility of the PacBio sequencing in rare disease diagnostics is amply demonstrated by showing the association of repeat expansions and acquisition of DNA methylation locally.”

- We thank the reviewer for this positive feedback.

“The central message of the paper talks about ‘hypermethylation outliers’, but this is extraordinarily opaque to the reader. This text really doesn’t convey enough information “In addition, we extracted the top 500 most variable autosomal CpGs based on a subset of the HiFi-GS samples sequenced at high depth (N=139) for CpG-based correlations. Similarly, we noted high correlations compared to randomly permuted values (median Pearson R=0.86, Extended Data 1)”, while the Methods section Classification of extreme hypermethylation outliers is likewise very difficult to follow – I have no idea how I would reproduce their analytical approach. Getting lost at this early stage creates problems for the rest of the manuscript. It’s not clear what is being compared to get the results of Extended Data 1, or whether the permutation approach is appropriate.”

- We have completely reformatted the paper and expanded the description and interpretation of all presented data for increased clarity.

“Another problem is the way the DNA methylation data are illustrated in the figures. The wiggle tracks of Extended Data 2 are clear, but when we start looking at the IGV screenshots of Extended Data 3-19, the impression is that the red (methylated) sites are not only more strikingly obvious, it looks like the number of CGs with calls is decreased when the site is unmethylated – the data look faded and sparse. As a result, we’re asked to believe that there is a change in DNA methylation when the figures look more like a failure to assign a methylation state. When these graphical outputs are compared with Figure 4 of Ni et al. (<https://www.biorxiv.org/content/10.1101/2022.02.26.482074v1.full>), the difference is striking. This discrepancy is unexplained and really limits the confidence in the results obtained.”

- As shown by technical comparisons of methylation scores, the presented data is accurate and not different to other published works (except our dataset is much larger in all aspects). While visualization differences with cited reference are unrelated, for clarity, we agree to utilize more dichotomous coloring scheme (offered by IGV 2.15.2) to highlight allelic and non-allelic methylation variation captured in the data. **Consequently, all figures are updated accordingly.**

“Figure 1 is likewise much too difficult to understand. Despite spending a significant amount of time trying to make sense of it I remain unable to explain what is being conveyed.”

- We have split this figure into separate presentations. First, in the newly added **Figure 3**, we present a simplified schematic depicting the process to identify rare hypermethylation outliers. Then, in the newly added **Figure 4**, we present genetic causes of rare hypermethylation events with improved description of each object.

“Figure 2 sounds like it is showing something valuable, a candidate regulatory variant at genes causing rare disease, but panel (a) suffers from unlabelled wiggle tracks and the strange faded depiction of what is described as unmethylated DNA, while panel (b) unfortunately failed to expand the gene annotation track, making it difficult to appreciate that the event is actually promoter-proximal. The complete lack of labelling for panel (c) is a major problem.”

- We apologize for unclear labeling of this important figure which we now have significantly improved. Specifically, the previously submitted Figure 2 is now Figure 6 and has, as indicated above, been reformatted with the new coloring scheme and improved labeling along with detailed description of the results. We discuss these findings in a separate paragraph **“Rare hypermethylation outlier facilitates diagnosis of unsolved rare disease cases” on Pages 11-12.**

Reviewer #2 (Remarks to the Author):

“The authors present work using PacBio HiFi sequencing to create haplotype-resolved data from a cohort of 276 samples / 152 families with rare diseases to identify rare hyper methylation events. A large number of events were allele-specific, and the authors demonstrated heritability. The authors identified an average of 30-40 hyper cpGs per individual that overlapped rare disease genes—more than I would have expected, but perhaps not surprising given the definition of a tile they used and the relaxed criteria for a hyper cpG. In general, I find the paper interesting and an important contribution to the field but with results that will need to be investigated further and validated by other studies. The methods allow for understanding of how the authors did the work for those familiar with long-read sequencing tools, but likely will be difficult for others to reproduce. I would encourage the editor to ask for additional data to be released to make the study replicable (see below). Besides data availability, I have no major concerns, just minor comments.”

- We are pleased to see that Reviewer#2 finds our work interesting and thank them for the careful review. As indicated above, we now provide additional data of the identified hyper-mCpG tiles in **Supplementary Table 4, 6 and 8.**

“The authors relaxed the definition of rare hyper cpG site in order to increase the number of candidate loci from 25,543 to an additional 31,726 sites. If they had not done this, would this have affected any of the major findings from the results? For example, would the DIP2B result have been flagged?”

- This is a great point by the Reviewer. We used two different approaches for identifying outliers: First, by restricting overall dispersion of methylation data (i.e., forcing z-score distribution to be narrow with max 2 outliers we captured extreme hyper-mCpG tiles that has no “natural” variation in methylation across individuals (i.e., static methylation level) which resulted in ~25,000 tiles. Then, secondly, by allowing wider overall z-score distribution (e.g., multiple unrelated samples with z-score >2), but extracting unusual observations among population variable (i.e., dynamic mCpG tiles) with z-score >5, minimum separation of 3 z-score units between max 2 z-scores and remainder of distribution we identified an additional set of ~31,000 unique extreme hyper-mCpG tiles. In case of the reported diagnostic finding of *DIP2B*, both approaches identify the extreme hyper-mCpG tiles. However, for some positive controls such as *DMPK* hypermethylation associated with diagnostic repeat expansion, the ‘natural’ methylation variation in population is substantial and only the 2nd approach identifies extreme hyper-mCpG tiles (z-score >15) clearly deviating from overall (wide) z-score distribution in population. We note that with increasing sample size (addressing both genetically and non-genetically

variable methylation in population) as well as conditioning outlier search with known common methylation variation (addressing pervasive genetic effects in common haplotypes, i.e., cis-mQTLs) it is likely that a single cut-off can be derived, which is a topic of our current ongoing work (but outside the scope of the manuscript here). **We have added clarified motivation for these two approaches in the newly added Results paragraph “Identification and characterization of rare hypermethylation outlier event in rare disease cases” on Pages 6-7 as well present a simplified schematic as Figure 3.**

“How many of these hyper-cpg tiles were within annotated transposable elements? Is there a significant enrichment for LINES? I’ve seen increases in local methylation, similar to the tiles the authors use, in these but have never managed to convince myself it correlates with anything meaningful.”

- To explore this interesting point raised by the Reviewer, we intersected the two lists of extreme hyper-mCpG tiles with the 'RepeatMasker' table available in the UCSC genome browser. By comparing with equal number of control regions (i.e., 200bp window mapping 10kb from each tile), we noted a slight enrichment of LINES for both lists (1.08 and 1.13-fold-enrichment for set 1 and set 2, respectively, Chi^2 $p < 0.0001$). Taken together, the minor (i.e., 8-13%) excess of LINE mapped to our outliers, while statistically significant, it accounts for small proportion of the data and with low enrichment we decided that discussion on this in current manuscript is not relevant.

Was the number of hyper cpg sites higher, lower, or similar in the 141 individuals with a known genetic diagnosis?

- The number of extreme hyper-mCpGs tiles per individual did not differ among unaffected and affected after taking sequencing coverage into account. **We have now added this important point to the manuscript on page 7.**

“I may just be missing this, but how did the authors handle repetitive regions where short reads align poorly? Were these excluded? Segmental duplications, for example.”

- As we have shown before (Cohen et al, Genet Med. 2022), over 80% of variant detection gain (i.e., unique non-singleton variants) by HiFi-GS in nearly 500 samples as compared to short-read based resources (e.g., gnomAD) maps to repetitive regions including segmental duplications. We use all aligned reads here for methylation and sequence variant detection regardless of their mapped location including segmental duplications and other repeat categories.

“Did the authors consider controlling for age when looking for hyper-cpg sites? We see age-related differences in our data, so I suspect it is in yours as well.”

- As shown by Seeboth et al (Clinical Epigenetics 2020), CpG methylation outlier burden appears to positively correlate with age even after correction for blood cell composition. While this finding agrees with the reviewer, we note that this linear relationship was

assessed in a large cohort of > 20,000 individuals aged 18 to 98 years. As our HiFi-GS data set presented here of high-covered individuals (i.e., probands) have a narrow age-range (2 to 10 years), we are unable to confirm a similar linear relationship and thus are not controlling for age.

“In all figures it might be difficult for readers not familiar with IGV and the phased view to discern the haplotypes. I would encourage the authors to add some kind of label to the haplotypes that makes them more clear beyond the small 1 and 2 from IGV.”

- As noted above in response to Reviewer#1, we have updated all main and supplementary figures for improved visualization.

“I found the IGV screenshot in figure 3C difficult to read as there is a lot of data—specifically, I can’t see how big those inserts are.”

- As noted above in response to Reviewer#1, we have updated all main and supplementary figures for improved visualization.

“In the spirit of data sharing and also making life easier for everyone I would encourage the authors to provide the tile windows/coordinates along with the average methylation for that tile in this population. Others could use this data as a comparator (myself included) for our own studies.”

- As noted above, we are now not only providing the tile coordinates but also average methylation rates for combined reads as well as separated for each haplotype for each hyper-mCpG tile. These are presented in **Supplementary Table 4; Supplementary Table 6 and Supplementary Table 8**.

REVIEWER COMMENTS

Reviewer #1 (Remarks to the Author):

In this revised manuscript, Cheung and colleagues have provided a number of updates in response to the prior reviews. My concerns had to do with the lack of clarity of the text and the accompanying figures, the current version has gone to significant lengths to address these concerns.

The first set of results shows concordance between DNA methylation values at specific loci using bisulphite sequencing compared with HiFi 5-base sequencing.

Secondly, they show how DNA methylation and repeat expansions can be observed on the same strand of DNA, in contrast to bisulphite sequencing. These data are convincing and will be clinically valuable.

Four supplementary figures are shown to demonstrate the value of the HiFi sequencing “for the diagnosis of imprinting disorders”. These results are confusing. Supplementary Figure 2 points us to “maternal allele-specific hypermethylation in proband” as depicted in black boxes, but what the boxes appear to be showing is a sequence variant (grey vertical line), with two of the three showing what looks like variants at CG dinucleotides. I am guessing that the authors mean to have us look at the black boxes to show that these depict the maternal allele, and that we should not be looking at those loci for maternal allele-specific hypermethylation, but rather at the flanking region. In addition, the way the text brings us into this set of results a reader may be expecting to see an actual disease state being portrayed, whereas we instead appear to be looking at a normal, physiological differentially-methylated region. Supplementary figures 3-5 are similarly misleading. Edits to allow a reader unfamiliar with the technology and IGV representations of these data to understand exactly what the authors are trying to communicate would make this set of results much more powerful.

It's also odd how the authors describe throughout the legends to figures how to interpret the colours: “...blue indicating low CpG methylation prediction and red indicating high CpG methylation prediction..” – an individual cytosine can't have low or high methylation, it can be unmethylated or methylated, this could be re-worded for accuracy.

The next section introduces “loss of RE (LRE)” – this abbreviation is introduced in the Abstract more completely as loss of regulatory element activity. Loss of regulatory element sounds like the sequence is deleted, whereas loss of regulatory element activity sounds like a more accurate description of what the authors are trying to convey. Retaining the fuller phrase (abbreviating to REA, LREA) would be a better idea.

Focusing on “extreme hyper-mCpG outliers” will certainly help to identify methylation quantitative trait loci (meQTLs), but the assumption is made that meQTLs always cause the unusual acquisition of DNA methylation, which is not justified and should be discussed.

In general, this section is improved compared with the original submission, although it could have benefited from more careful proof-reading. Figure 3 illustrates the text, but both remain difficult to understand.

Results are shown in Figure 4 and show an enrichment for variants in cis compared with in trans. Missing from the figure is what's described as "E. Genomic view of an example of a rare SNV (black box) mapping in cis close to a hyper-mCpG tile on chromosome 10 causing allele-specific hypermethylation. Track depicts haplotype-resolved HiFi-GS reads with CpG modification staining (blue indicating low methylation prediction and red indicating high methylation prediction). Hap 1 denotes haplotype 1 and Hap 2 denotes haplotype 2." This seems to have ended up as Supplementary Figure 6. This will only make sense to a reader in the context of showing a separate sample lacking the variant in which the DNA methylation is not changed.

I'm a bit concerned about assertions of causality. For example, in the legend for Supplementary figure 5, it is stated that "Genomic view of an example of a rare SNV (black box) mapping in cis close to a hyper-mCpG tile on chromosome 22 causing allele-specific hypermethylation." It's probably causing the DNA methylation change, but having just showed in Figure 4 that DNA methylation changes from SNVs can be 1 kb away, how can we be sure that this SNV in this 490 bp window is causal? It's an association at best based on the evidence presented, there could be another variant in the same haplotype that is causal.

I don't understand what is meant to be conveyed in Supplementary Figure 10.

The mapping of "extreme hyper-mCpG tiles" to regulatory elements in Figure 5a lacks any way of interpreting the significance of the overlap. Statistical testing should be performed to allow this finding to be understood, ideally focusing on regulatory elements mapped in the same cell/tissue type.

The results described as "Validation of rare hypermethylation events by isoform profiling" do not appear to describe different isoforms, but instead allelic imbalance.

The final set of results show some interesting associations of meQTLs with disease genes, but with minimal clinical information that would allow the reader to understand potential causality.

I note the clarifications in the Methods section.

I appreciate that the authors have been responsive in performing numerous edits to address my prior concerns. Now that the manuscript is easier to interpret, new problems have become apparent. There is still room for substantial editing for clarity, there are mistakes in describing figure components that are not present, relatively minor but necessary edits. What is more worrisome is that the results are generally superficial and descriptive but are presented as if the candidate meQTLs are definitively causal of the DNA methylation changes. For the local effects at repeat expansions, that's pretty reasonable to assert, but for the other examples what we're seeing is interesting associations, findings that would not have been possible with short read sequencing and warrant further testing, but it's too much to say that all these variants are causal.

Reviewer #2 (Remarks to the Author):

The authors have addressed my major concerns with the manuscript. I have no additional concerns.

AUTHOR SUMMARY

We are pleased to see the mutual enthusiasm from both Reviewers about our revision.

- Reviewer #1 states that “..the current version has gone to significant lengths to address these concerns.”, “..data are convincing and will be clinically valuable.”, “I appreciate that the authors have been responsive in performing numerous edits to address my prior concerns.”
- Reviewer #2 has no further request as indicated by the statement: “The authors have addressed my major concerns with the manuscript. I have no additional concerns.”

We also thank them again for their careful second review and address the additional comments raised by Reviewer#1, as outlined further below.

REVIEWER COMMENTS AND AUTHOR RESPONSE

Reviewer #1 (Remarks to the Author):

“In this revised manuscript, Cheung and colleagues have provided a number of updates in response to the prior reviews. My concerns had to do with the lack of clarity of the text and the accompanying figures, the current version has gone to significant lengths to address these concerns. The first set of results shows concordance between DNA methylation values at specific loci using bisulphite sequencing compared with HiFi 5-base sequencing. Secondly, they show how DNA methylation and repeat expansions can be observed on the same strand of DNA, in contrast to bisulphite sequencing. These data are convincing and will be clinically valuable.”

- We thank the reviewer again for their careful review and are pleased to see that our extensive revision was to large extent satisfactory and the agreement that the data will be clinically valuable.

“Four supplementary figures are shown to demonstrate the value of the HiFi sequencing “for the diagnosis of imprinting disorders”. These results are confusing. Supplementary Figure 2 points us to “maternal allele-specific hypermethylation in proband” as depicted in black boxes, but what the boxes appear to be showing is a sequence variant (grey vertical line), with two of the three showing what looks like variants at CG dinucleotides. I am guessing that the authors mean to have us look at the black boxes to show that these depict the maternal allele, and that we should not be looking at those loci for maternal allele-specific hypermethylation, but rather at the flanking region. In addition, the way the text brings us into this set of results a reader may be expecting to see an actual disease state being portrayed, whereas we instead appear to be looking at a normal, physiological differentially-methylated region. Supplementary figures 3-5 are similarly misleading. Edits to allow a reader unfamiliar with the technology and IGV representations of these data to understand exactly what the authors are trying to communicate would make this set of results much more powerful.”

- We apologize for the confusion about the utility of 5-base HiFi-GS for detecting imprinting disorders. We have clarified in the text that we use unaffected individuals to resolve the parent of origin of the allele-specific methylation pattern at causal regions linked to different imprinting disorders. These changes can be found on Page 6. We have also updated the legends for Supplementary Figure 2-5 for improved clarity.

“It's also odd how the authors describe throughout the legends to figures how to interpret the colours: “...blue indicating low CpG methylation prediction and red indicating high CpG methylation prediction..” – an individual cytosine can't have low or high methylation, it can be unmethylated or methylated, this could be re-worded for accuracy.”

- The reviewer is right, in native DNA a cytosine can either be unmodified or methylated to 5-methylcytosine (5mC). However, the HiFi-GS methylation calling algorithm is different from methods utilizing sodium bisulfite conversion (WGBS, Illumina EPIC/450K array etc.) for single base resolution. Instead, 5-base HiFi-GS uses a deep learning model that integrates sequencing kinetics and base context for methylation probability without requiring bisulfite conversion. Specifically, epigenetic modifications such as 5mC impacts polymerase kinetics (i.e., how fast bases are incorporated) and a convolutional neural network model that runs directly on the Sequel IIe system processes this polymerase kinetics and determine the probability of methylation status (unmethylated or methylated) of each CpG site in a HiFi read. However, we have re-worded the explanation of the red/blue coloring in figure legends to “Haplotype-resolved HiFi-GS reads with CpG modification coloring (blue indicating unmethylated CpG prediction and red indicating methylated CpG prediction with color intensity corresponding to base modification probabilities) throughout main and supplementary figures, respectively.

“The next section introduces “loss of RE (LRE)” – this abbreviation is introduced in the Abstract more completely as loss of regulatory element activity. Loss of regulatory element sounds like the sequence is deleted, whereas loss of regulatory element activity sounds like a more accurate description of what the authors are trying to convey. Retaining the fuller phrase (abbreviating to REA, LREA) would be a better idea.”

- We thank the reviewer for this suggestion which we have adopted. These changes can be found in Abstract (Page 2), Introduction (Page 5) and Results (Page 7) section, respectively.

“Focusing on “extreme hyper-mCpG outliers” will certainly help to identify methylation quantitative trait loci (meQTLs), but the assumption is made that meQTLs always cause the unusual acquisition of DNA methylation, which is not justified and should be discussed.”

- While we note in Figure 4 that local rare *cis*-variants are enriched near methylation outliers, similar to gene expression outliers (*Fresard et al Nature Medicine 2019*), these outliers can have multiple sources (environmental or complex / distal genetic causes). However, in contrast to expression traits, most regulatory sequences can be linked to informative heterozygous variants in long 5mC-HiFiGS reads. This haploid specificity allowed us to focus on ~15% of outliers in proximity of rare variants in same chromosome. We have amended the discussion accordingly on Page 13.

“In general, this section is improved compared with the original submission, although it could have benefited from more careful proof-reading. Figure 3 illustrates the text, but both remain difficult to understand.”

- We have made some additional changes to Figure 3 to further clarify the process to identify rare hypermethylation outliers in our cohort.

“Results are shown in Figure 4 and show an enrichment for variants in cis compared with in trans. Missing from the figure is what’s described as “E. Genomic view of an example of a rare SNV (black box) mapping in cis close to a hyper-mCpG tile on chromosome 10 causing allele-specific hypermethylation. Track depicts haplotype-resolved HiFi-GS reads with CpG modification staining (blue indicating low methylation prediction and red indicating high methylation prediction). Hap 1 denotes haplotype 1 and Hap 2 denotes haplotype 2.” This seems to have ended up as Supplementary Figure 6. This will only make sense to a reader in the context of showing a separate sample lacking the variant in which the DNA methylation is not changed.”

- We sincerely apologize for this typo as the intention was to have it as Supplementary Figure 6 which has now been corrected. In addition, as suggested by the reviewer, we have now added a separate sample lacking the variant to further exemplify the effect on CpG methylation by the rare allele.

“I’m a bit concerned about assertions of causality. For example, in the legend for Supplementary figure 5, it is stated that “Genomic view of an example of a rare SNV (black box) mapping in cis close to a hyper-mCpG tile on chromosome 22 causing allele-specific hypermethylation.” It’s probably causing the DNA methylation change, but having just showed in Figure 4 that DNA methylation changes from SNVs can be 1 kb away, how can we be sure that this SNV in this 490 bp window is causal? It’s an association at best based on the evidence presented, there could be another variant in the same haplotype that is causal.”

- While we show in Figure 4D that the strongest enrichment for cis-acting rare SNVs is within 200bp we have amended the text to refer to an association, as suggested by the reviewer. These changes can be found in the Abstract (Page 2), Figure 4 Legend (Page 33), Supplementary Figure 7 Legend, Supplementary Figure 8 Legend, Supplementary Figure 13 Legend, Supplementary Figure 14 Legend, Supplementary Figure 15 Legend, Supplementary Figure 16 Legend, Supplementary Figure 17 Legend, Supplementary Figure 18 Legend.

“I don’t understand what is meant to be conveyed in Supplementary Figure 10.”

- As presented in the Results section (Page 9) we note that most hypermethylation outlier effects are restricted to a few hundred bp corresponding to two or less hyper-mCpG tiles which we refer to as ‘regular’ hyper-mCpG tiles. We exemplify this in Supplementary Fig. 10 by assessing the significance of differential methylation per CpG using 2-by-2 Fisher’s exact test examining each CpG state (methylated vs. unmethylated) in reads with rare C versus common A-allele carrying reads, respectively. The red box shows the region. However, we have updated the figure to show the different alleles more clearly highlighting the A to C transversion and the association to hypermethylation.

“The mapping of “extreme hyper-mCpG tiles” to regulatory elements in Figure 5a lacks any way of interpreting the significance of the overlap. Statistical testing should be performed to allow this finding to be understood, ideally focusing on regulatory elements mapped in the same cell/tissue type.”

- We have now performed statistical testing showing the significant enrichment of rare hyper-mCpG tiles across multiple cell types. The changes can be found in the Results section (Page 9) and in Figure 5a.

“The results described as “Validation of rare hypermethylation events by isoform profiling” do not appear to describe different isoforms, but instead allelic imbalance.”

- We thank the reviewer for this important point which we have corrected. The title of this paragraph now reads: “Validation of rare hypermethylation events by full-length cDNA sequencing”. These changes can be found on Page 10.

“The final set of results show some interesting associations of meQTLs with disease genes, but with minimal clinical information that would allow the reader to understand potential causality. “

- We have amended the Results section on Page 12 to add more clinical information of the two genes that were followed-up like *GNAO1* and *DIP2B* as follows: “The hyper-mCpG events at the 5’ proximal promoters of the dominant disease genes NSD1 and SET were identified in two patients whose clinical features were not compatible with the syndromes linked to loss-of-function in these loci.”

“I note the clarifications in the Methods section. I appreciate that the authors have been responsive in performing numerous edits to address my prior concerns. Now that the manuscript is easier to interpret, new problems have become apparent. There is still room for substantial editing for clarity, there are mistakes in describing figure components that are not present, relatively minor but necessary edits. What is more worrisome is that the results are generally superficial and descriptive but are presented as if the candidate meQTLs are definitively causal of the DNA methylation changes. For the local effects at repeat expansions, that’s pretty reasonable to assert, but for the other examples what we’re seeing is interesting associations, findings that would not have been possible with short read sequencing and warrant further testing, but it’s too much to say that all these variants are causal.”

- We are pleased to see that the Reviewer appreciates our revision. As indicated above, we have made further clarifications and edits based on the specific comments.

Reviewer #2 (Remarks to the Author):

“The authors have addressed my major concerns with the manuscript. I have no additional concerns.”

- We thank the reviewer again for their careful review and are pleased to see that our revision was satisfactory.

REVIEWERS' COMMENTS

Reviewer #1 (Remarks to the Author):

The authors have addressed my specific issues and have created a very informative and interesting report.